# A Holistic Approach to Unifying Automatic Concept Extraction and Concept Importance Estimation

**Thomas Fel**[*,1,2], **Victor Boutin**[*1,2],
**Mazda Moayeri**[3], **Rémi Cadène**[1], **Louis Bethune**[2] **Léo Andéol**[2], **Mathieu Chalvidal**[1,2],
**Thomas Serre**[1,2]
[1]Carney Institute for Brain Science, Brown University
[2]Artificial and Natural Intelligence Toulouse Institute
[3]Department of Computer Science, University of Maryland
{thomas_fel,victor_boutin}@brown.edu

## Abstract

In recent years, concept-based approaches have emerged as some of the most promising explainability methods to help us interpret the decisions of Artificial Neural Networks (ANNs). These methods seek to discover intelligible visual "concepts" buried within the complex patterns of ANN activations in two key steps: (1) concept extraction followed by (2) importance estimation. While these two steps are shared across methods, they all differ in their specific implementations. Here, we introduce a unifying theoretical framework that recast the first step – concept extraction problem – as a special case of **dictionary learning**, and we formalize the second step – concept importance estimation – as a more general form of **attribution method**. This framework offers several advantages as it allows us: *(i)* to propose new evaluation metrics for comparing different concept extraction approaches; *(ii)* to leverage modern attribution methods and evaluation metrics to extend and systematically evaluate state-of-the-art concept-based approaches and importance estimation techniques; *(iii)* to derive theoretical guarantees regarding the optimality of such methods.

We further leverage our framework to try to tackle a crucial question in explainability: how to *efficiently* identify clusters of data points that are classified based on a similar shared strategy. To illustrate these findings and to highlight the main strategies of a model, we introduce a visual representation called the strategic cluster graph. Finally, we present 🔎 Lens, a dedicated website that offers a complete compilation of these visualizations for all classes of the ImageNet dataset.

## 1 Introduction

The black-box nature of Artificial Neural Networks (ANNs) poses a significant hurdle to their deployment in industries that must comply with stringent ethical and regulatory standards [1]. In response to this challenge, eXplainable Artificial Intelligence (XAI) focuses on developing new tools to help humans better understand how ANNs arrive at their decisions [2, 3]. Among the large array of methods available, attribution methods have become the go-to approach [4–14]. They yield heatmaps in order to highlight the importance of each input feature (or group of features [15]) for driving a model's decision. However, there is growing consensus that these attribution methods fall short of providing meaningful explanations [16–19] as revealed by multiple user studies [20–25, 20]. It has been suggested that for explainability methods to become usable by human users, they need to be able to highlight not just the location of important features within an image (i.e., the *where* information) but also their semantic content (i.e., the *what* information).

---

[*] The authors contributed equally.

37th Conference on Neural Information Processing Systems (NeurIPS 2023).

One promising set of explainability methods to address this challenge includes concept-based explainability methods, which are methods that aim to identify high-level concepts within the activation space of ANNs [26]. These methods have recently gained renewed interest due to their success in providing human-interpretable explanations [27–30] (see section 2 for a detailed description of the related work). However, concept-based explainability methods are still in the early stages, and progress relies largely on researchers' intuitions rather than well-established theoretical foundations. A key challenge lies in formalizing the notion of concept itself [31]. Researchers have proposed desiderata such as meaningfulness, coherence, and importance [27] but the lack of formalism in concept definition has hindered the derivation of appropriate metrics for comparing different methods.

This article presents a theoretical framework to unify and characterize current concept-based explainability methods. Our approach builds on the fundamental observation that all concept-based explainability methods share two key steps: (1) concepts are extracted, and (2) importance scores are assigned to these concepts based on their contribution to the model's decision [27]. Here, we show how the first extraction step can be formulated as a dictionary learning problem while the second importance scoring step can be formulated as an attribution problem in the concept space. To summarize, our contributions are as follows:

- We describe a novel framework that unifies all modern concept-based explainability methods and we borrow metrics from different fields (such as sparsity, reconstruction, stability, FID, or OOD scores) to evaluate the effectiveness of those methods.
- We leverage modern attribution methods to derive seven novel concept importance estimation methods and provide theoretical guarantees regarding their optimality. Additionally, we show how standard faithfulness evaluation metrics used to evaluate attribution methods (i.e., Insertion, Deletion [32], and $\mu$Fidelity [33]) can be adapted to serve as benchmarks for concept importance scoring. In particular, we demonstrate that Integrated Gradients, Gradient Input, RISE, and Occlusion achieve the highest theoretical scores for 3 faithfulness metrics when the concept decomposition is on the penultimate layer.
- We introduce the notion of local concept importance to address a significant challenge in explainability: the identification of image clusters that reflect a shared strategy by the model (see Figure 1). We show how the corresponding cluster plots can be used as visualization tools to help with the identification of the main visual strategies used by a model to help explain false positive classifications.

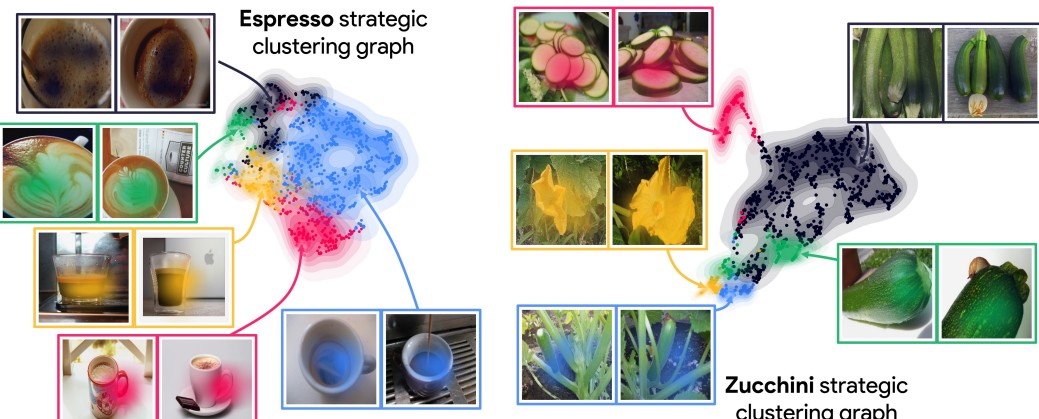

Figure 1: **Strategic cluster graphs for the espresso and zucchini classes.** The framework presented in this study provides a comprehensive approach to uncover local importance using any attribution methods. Consequently, it allow us to estimate the critical concepts influencing the model's decision for each image. As a results, we introduced the Strategic cluster graph, which offers a visual representation of the main strategies employed by the model in recognizing an entire object class. For espresso (left), the main strategies for classification appear to be: ● bubbles and foam on the coffee, ● Latte art, ● transparent cups with foam and black liquid, ● the handle of the coffee cup, and finally ● the coffee in the cup, which appears to be the predominant strategy. As for zucchini, the strategies are: ● a zucchini in a vegetable garden, ● the corolla of the zucchini flower, ● sliced zucchini, ● the spotted pattern on the zucchini skin and ● stacked zucchini.

## 2  Related Work

Kim *et al.* [26] were the first to propose a concept-based approach to interpret neural network internal states. They defined the notion of concepts using Concept Activation Vectors (CAVs). CAVs are derived by training a linear classifier between a concept's examples and random counterexamples and then taking the vector orthogonal to the decision boundary. In their work, the concepts are manually selected by humans. They further introduce the first concept importance scoring method, called Testing with CAVs (TCAV). TCAV uses directional derivatives to evaluate the contribution of each CAV to the model's prediction for each object category. Although this approach demonstrates meaningful explanations to human users, it requires a significant human effort to create a relevant image database of concepts. To address this limitation, Ghorbani *et al.* [27] developed an unsupervised method called Automatic Concept Extraction (ACE) that extracts CAVs without the need for human supervision. In their work, the CAVs are the centroid of the activations (in a given layer) when the network is fed with multi-scale image segments belonging to an image class of interest. However, the use of image segments could introduce biases in the explanations [34–37]. ACE also leverages TCAV to rank the concepts of a given object category based on their importance.

Zhang *et al.* [28] proposed a novel method for concept-based explainability called Invertible Concept-based Explanation (ICE). ICE leverages matrix factorization techniques, such as non-negative matrix factorization (NMF), to extract Concept Activation Vectors (CAVs). Here, the concepts are localized as the matrix factorization is applied on feature maps (before the global average pooling). In ICE, the concepts' importance is computed using the TCAV score [26]. Note that the Singular Value Decomposition (SVD) was also suggested as a concept discovery method [28, 30]. CRAFT (Concept Recursive Activation FacTorization for explainability) uses NMF to extract the concepts, but as it is applied after the global pooling average, the concepts are location invariant. Additionally, CRAFT employs Sobol indices to quantify the global importance of concepts associated with an object category.

## 3  A Unifying perspective

**Notations.** Throughout, $|| \cdot ||_2$ and $|| \cdot ||_F$ represent the $\ell_2$ and Frobenius norm, respectively. We consider a general supervised learning setting, where a classifier $\boldsymbol{f} : \mathcal{X} \rightarrow \mathcal{Y}$ maps inputs from an input space $\mathcal{X} \subseteq \mathbb{R}^d$ to an output space $\mathcal{Y} \subseteq \mathbb{R}^c$. For any matrix $\mathbf{X} \in \mathbb{R}^{n \times d}$, $\boldsymbol{x}_i$ denotes the $i^{th}$ row of $\mathbf{X}$, where $i \in \{1, \dots, n\}$ and $\boldsymbol{x}_i \in \mathbb{R}^d$. Without loss of generality, we assume that $\boldsymbol{f}$ admits an intermediate space $\mathcal{H} \subseteq \mathbb{R}^p$. In this setup, $\boldsymbol{h} : \mathcal{X} \rightarrow \mathcal{H}$ maps inputs to the intermediate space, and $\boldsymbol{g} : \mathcal{H} \rightarrow \mathcal{Y}$ takes the intermediate space to the output. Consequently, $\boldsymbol{f}(\boldsymbol{x}) = (\boldsymbol{g} \circ \boldsymbol{h})(\boldsymbol{x})$. Additionally, let $\boldsymbol{a} = \boldsymbol{h}(\boldsymbol{x}) \in \mathcal{H}$ represent the activations of $\boldsymbol{x}$ in this intermediate space. We also abuse notation slightly: $\boldsymbol{f}(\mathbf{X}) = (\boldsymbol{g} \circ \boldsymbol{h})(\mathbf{X})$ denotes the vectorized application of $\boldsymbol{f}$ on each element $\boldsymbol{x}$ of $\mathbf{X}$, resulting in $(\boldsymbol{f}(\boldsymbol{x}_1), \dots, \boldsymbol{f}(\boldsymbol{x}_n))$.

Prior methods for concept extraction, namely ACE [27], ICE [28] and CRAFT [29], can be distilled into two fundamental steps:

(*i*) **Concept extraction:** A set of images $\mathbf{X} \in \mathbb{R}^{n \times d}$ belonging to the same class is sent to the intermediate space giving activations $\mathbf{A} = \boldsymbol{h}(\mathbf{X}) \in \mathbb{R}^{n \times p}$. These activations are used to extract a set of $k$ CAVs using K-Means [27], PCA (or SVD) [28, 30] or NMF [28, 29]. Each CAV is denoted $\boldsymbol{v}_i$ and $\mathbf{V} = (\boldsymbol{v}_1, \dots, \boldsymbol{v}_k) \in \mathbb{R}^{p \times k}$ forms the dictionary of concepts.

(*ii*) **Concept importance scoring:** It involves calculating a set of $k$ global scores, which provides an importance measure of each concept $\boldsymbol{v}_i$ to the class as a whole. Specifically, it quantifies the influence of each concept $\boldsymbol{v}_i$ on the final classifier prediction for the given set of points $\mathbf{X}$. Prominent measures for concept importance include TCAV [26] and the Sobol indices [29].

The two-step process described above is repeated for all classes. In the following subsections, we theoretically demonstrate that the concept extraction step *(i)* could be recast as a dictionary learning problem (see 3.1). It allows us to reformulate and generalize the concept importance step *(ii)* using attribution methods (see 3.2).

### 3.1 Concept Extraction

**A dictionary learning perspective.** The purpose of this section is to redefine all current concept extraction methods as a problem within the framework of dictionary learning. Given the necessity for clearer formalization and metrics in the field of concept extraction, integrating concept extraction with dictionary learning enables us to employ a comprehensive set of metrics and obtain valuable theoretical insights from a well-established and extensively researched domain.

The goal of concept extraction is to find a small set of interpretable CAVs (i.e., $\mathbf{V}$) that allows us to faithfully interpret the activation $\mathbf{A}$. By preserving a linear relationship between $\mathbf{V}$ and $\mathbf{A}$, we facilitate the understanding and interpretability of the learned concepts [26, 38]. Therefore, we look for a coefficient matrix $\mathbf{U} \in \mathbb{R}^{n \times k}$ (also called loading matrix) and a set of CAVs $\mathbf{V}$, so that $\mathbf{A} \approx \mathbf{U}\mathbf{V}^{\mathsf{T}}$. In this approximation of $\mathbf{A}$ using the two low-rank matrices $(\mathbf{U}, \mathbf{V})$, $\mathbf{V}$ represents the concept basis used to reinterpret our samples, and $\mathbf{U}$ are the coordinates of the activation in this new basis. Interestingly, such a formulation allows a recast of the concept extraction problem as an instance of dictionary learning problem [39] in which all known concept-based explainability methods fall:

$$(\mathbf{U}^{\star}, \mathbf{V}^{\star}) = \underset{\mathbf{U}, \mathbf{V}}{\arg\min} ||\mathbf{A} - \mathbf{U}\mathbf{V}^{\mathsf{T}}||_F^2 \;\; s.t \begin{cases} \forall\, i, \boldsymbol{u}_i \in \{\mathbf{e}_1, \ldots, \mathbf{e}_k\} \;\; \text{(K-Means : \textbf{ACE} [27])}, & (1) \\ \mathbf{V}^{\mathsf{T}}\mathbf{V} = \mathbf{I} \;\; \text{(PCA: [28, 30])}, & (2) \\ \mathbf{U} \geq 0, \mathbf{V} \geq 0 \;\; \text{(NMF : \textbf{CRAFT} [29] \& \textbf{ICE} [28])} & (3) \\ \mathbf{U} = \boldsymbol{\psi}(\mathbf{A}), ||\mathbf{U}||_0 \leq K \;\; \text{(Sparse Autoencoder [40])} & (4) \end{cases}$$

with $\boldsymbol{e}_i$ the $i$-th element of the canonical basis, $\mathbf{I}$ the identity matrix and $\boldsymbol{\psi}$ any neural network. In this context, $\mathbf{V}$ is the *dictionary* and $\mathbf{U}$ the *representation* of $\mathbf{A}$ with the atoms of $\mathbf{V}$. $\boldsymbol{u}_i$ denote the $i$-th row of $\mathbf{U}$. These methods extract the concept banks $\mathbf{V}$ differently, thereby necessitating different interpretations[*].

In **ACE**, the CAVs are defined as the centroids of the clusters found by the K-means algorithm. Specifically, a concept vector $\boldsymbol{v}_i$ in the matrix $\mathbf{V}$ indicates a dense concentration of points associated with the corresponding concept, implying a repeated activation pattern. The main benefit of ACE comes from its reconstruction process, involving projecting activations onto the nearest centroid, which ensures that the representation will lie within the observed distribution (no out-of-distribution instances). However, its limitation lies in its lack of expressivity, as each activation representation is restricted to a single concept ($||\boldsymbol{u}||_0 = 1$). As a result, it cannot capture compositions of concepts, leading to sub-optimal representations that fail to fully grasp the richness of the underlying data distribution.

On the other hand, the PCA benefits from superior reconstruction performance due to its lower constraints, as stated by the Eckart-Young-Mirsky[43] theorem. The CAVs are the eigenvector of the covariance matrix: they indicate the direction in which the data variance is maximal. An inherent limitation is that the PCA will not be able to properly capture stable concepts that do not contribute to the sample variability (e.g. the dog-head concept might not be considered important by the PCA to explain the dog class if it is present across all examples). Neural networks are known to cluster together the points belonging to the same category in the last layer to achieve linear separability ([44, 29]). Thus, the orthogonality constraint in the PCA might not be suitable to correctly interpret the manifold of the deep layer induced by points from the same class (it is interesting to note that this limitation can be of interest when studying all classes at once). Also, unlike K-means, which produces strictly positive clusters if all points are positive (e.g., the output of ReLU), PCA has no sign constraint and can undesirably reconstruct out-of-distribution (OOD) activations, including negative values after ReLU.

In contrast to K-Means, which induces extremely sparse representations, and PCA, which generates dense representations, the NMF (used in **CRAFT** and **ICE**) strikes a harmonious balance as it provides moderately sparse representation. This is due to NMF relaxing the constraints imposed by the K-means algorithm (adding an orthogonality constraint on $\mathbf{V}$ such that $\mathbf{V}\mathbf{V}^{\mathsf{T}} = \mathbf{I}$ would yield an equivalent solution to K-means clustering [45]). This sparsity facilitates the encoding of

---

[*]Concept extractions are typically overcomplete dictionaries, meaning that if the dictionary for each class is combined, $k > p$, as noted in [29]. Recently, [29] and a more detailed work [41] suggest that overcomplete dictionaries are serious candidates to the superposition problem [42].

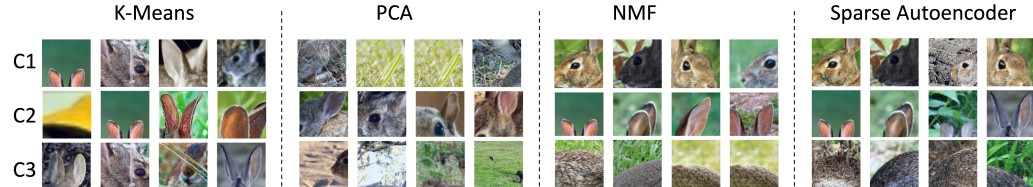

Figure 2: **Most important concepts extracted for the studied methods.** This qualitative example shows the three most important concepts extracted for the 'rabbit' class using a ResNet50 trained on ImageNet. The crops correspond to those maximizing each concepts $i$ (i.e., $\boldsymbol{x}$ where $\mathbf{U}(\boldsymbol{x})_i$ is maximal). As demonstrated in previous works [28, 29, 49], NMF (requiring positive activations) produces particularly interpretable concepts despite poorer reconstruction than PCA and being less sparse than K-Means. Details for the sparse Autoencoder architecture are provided in the appendix.

compositional representations that are particularly valuable when an image encompasses multiple concepts. Moreover, by allowing only additive linear combinations of components with non-negative coefficients, NMF inherently fosters a parts-based representation. This distinguishes NMF from PCA, which offers a holistic representation model. Interestingly, the NMF is known to yield representations that are interpretable by humans [28, 29]. Finally, the non-orthogonality of these concepts presents an advantage as it accommodates the phenomenon of superposition [38], wherein neurons within a layer may contribute to multiple distinct concepts simultaneously.

To summarize, we have explored three approaches to concept extraction, each necessitating a unique interpretation of the resulting Concept Activation Vectors (CAVs). Among these methods, NMF (used in **CRAFT** and **ICE**) emerges as a promising middle ground between PCA and K-means. Leveraging its capacity to capture intricate patterns, along with its ability to facilitate compositional representations and intuitive parts-based interpretations (as demonstrated in Figure 2), NMF stands out as a compelling choice for extracting meaningful concepts from high-dimensional data. These advantages have been underscored by previous human studies, as evidenced by works such as Zhang et al.[28] and Fel et al.[29].

|  | Relative $\ell_2$ ($\downarrow$) | Sparsity ($\uparrow$) | Stability ($\downarrow$) | FID ($\downarrow$) | OOD ($\downarrow$) |
|---|---|---|---|---|---|
|  | Eff / R50 / Mob | Eff / R50 / Mob | Eff / R50 / Mob | Eff / R50 / Mob | Eff / R50 / Mob |
| PCA | 0.60 / 0.54 / 0.73 | 0.00 / 0.00 / 0.0 | 0.41 / 0.38 / 0.43 | 0.47 / 0.17 / 0.24 | 2.44 / 0.36 / 0.16 |
| KMeans | 0.72 / 0.66 / 0.84 | 0.95 / 0.95 / 0.95 | 0.07 / 0.08 / 0.04 | 0.46 / 0.21 / 0.33 | 1.76 / 0.29 / 0.15 |
| NMF | 0.63 / 0.57 / 0.75 | 0.68 / 0.44 / 0.64 | 0.17 / 0.14 / 0.16 | 0.38 / 0.21 / 0.24 | 1.98 / 0.29 / 0.15 |

Table 1: **Concept extraction comparison.** Eff, R50 and Mob denote EfficientNetV2[46], ResNet50[47], MobileNetV2[48]. The concept extraction methods are applied on the last layer of the networks. Each results is averaged across 10 classes of ImageNet and obtained from a set of 16k images for each class.

**Evaluation of concept extraction** Following the theoretical discussion of the various concept extraction methods, we conduct an empirical investigation of the previously discussed properties to gain deeper insights into their distinctions and advantages. In our experiment, we apply the PCA, K-Means, and NMF concept extraction methods on the penultimate layer of three state-of-the-art models. We subsequently evaluate the concepts using five different metrics (see Table 1). All five metrics are connected with the desired characteristics of a dictionary learning method. They include achieving a high-quality reconstruction (Relative l2), sparse encoding of concepts (Sparsity), ensuring the stability of the concept base in relation to $\mathbf{A}$ (Stability), performing reconstructions within the intended domain (avoiding OOD), and maintaining the overall distribution during the reconstruction process (FID). All the results come from 10 classes of ImageNet (the one used in Imagenette [50]), and are obtained using $n = 16k$ images for each class.

We begin our empirical investigation by using a set of standard metrics derived from the dictionary learning literature, namely Relative $l_2$ and Sparsity. Concerning the Relative $\ell_2$, PCA achieves the highest score among the three considered methods, confirming the theoretical expectations based on the Eckart–Young–Mirsky theorem [43], followed by NMF. Concerning the sparsity of the underlying representation $\boldsymbol{u}$, we compute the proportion of non-zero elements $||\boldsymbol{u}||_0/k$. Since

K-means inherently has a sparsity of $1/k$ (as induced by equation 1), it naturally performs better in terms of sparsity, followed by NMF.

We deepen our investigation by proposing three additional metrics that offer complementary insights into the extracted concepts. Those metrics are the Stability, the FID, and the OOD score. The Stability (as it can be seen as a loose approximation of algorithmic stability [51]) measures how consistent concepts remain when they are extracted from different subsets of the data. To evaluate Stability, we perform the concept extraction methods $N$ times on $K$-fold subsets of the data. Then, we map the extracted concepts together using a Hungarian loss function and measure the cosine similarity of the CAVs. If a method is stable, it should yield the same concepts (up to permutation) across each $K$-fold, where each fold consists of 1000 images. K-Means and NMF demonstrate the highest stability, while PCA appears to be highly unstable, which can be problematic for interpreting the results and may undermine confidence in the extracted concepts.

The last two metrics, FID and OOD, are complementary in that they measure: (i) how faithful the representations extracted are w.r.t the original distribution, and (ii) the ability of the method to generate points lying in the data distribution (non-OOD). Formally, the FID quantifies the 1-Wasserstein distance [52] $\mathcal{W}_1$ between the empirical distribution of activation $\mathbf{A}$, denoted $\mu_{\boldsymbol{a}}$, and the empirical distribution of the reconstructed activation $\mathbf{U}\mathbf{V}^{\mathsf{T}}$ denoted $\mu_{\boldsymbol{u}}$. Thus, FID is calculated as $\text{FID} = \mathcal{W}_1(\mu_{\boldsymbol{a}}, \mu_{\boldsymbol{u}})$. On the other hand, the OOD score measures the plausibility of the reconstruction by leveraging Deep-KNN [53], a recent state-of-the-art OOD metric. More specifically, we use the Deep-KNN score to evaluate the deviation of a reconstructed point from the closest original point. In summary, a good reconstruction method is capable of accurately representing the original distribution (as indicated by FID) while ensuring that the generated points remain within the model's domain (non-OOD). K-means leads to the best OOD scores because each instance is reconstructed as a centroid, resulting in proximity to in-distribution (ID) instances. However, this approach collapses the distribution to a limited set of points, resulting in low FID. On the other hand, PCA may suffer from mapping to negative values, which can adversely affect the OOD score. Nevertheless, PCA is specifically optimized to achieve the best average reconstructions. NMF, with fewer stringent constraints, strikes a balance by providing in-distribution reconstructions at both the sample and population levels.

In conclusion, the results clearly demonstrate NMF as a method that strikes a balance between the two approaches as NMF demonstrates promising performance across all tested metrics. Henceforth, we will use the NMF to extract concepts without mentioning it.

**The Last Layer as a Promising Direction**    The various methods examined, namely **ACE**, **ICE**, and **CRAFT**, generally rely on a deep layer to perform their decomposition without providing quantitative or theoretical justifications for their choice. To explore the validity of this choice, we apply the aforementioned metrics to each block's output in a ResNet50 model. Figure 3 illustrates the metric evolution across different blocks, revealing a trend that favors the last layer for the decomposition. This empirical finding aligns with the practical implementations discussed above.

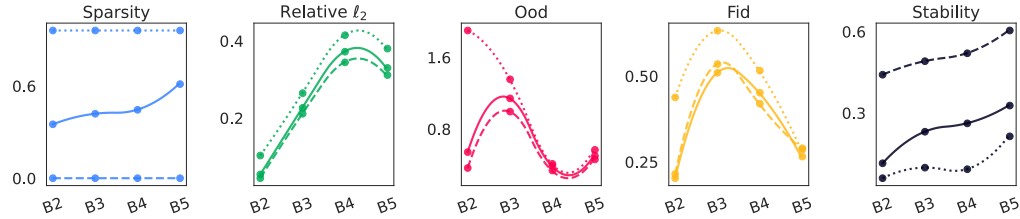

Figure 3: **Concept extraction metrics across layers.** The concept extraction methods are applied on activations probed on different blocks of a ResNet50 (B2 to B5). Each point is averaged over 10 classes of ImageNet using 16k images for each class. We evaluate 3 concept extraction methods: PCA (- - -), NMF (——), and KMeans (······).

### 3.2   Concept importance

In this section, we leverage our framework to unify concept importance scoring using the existing attribution methods. Furthermore, we demonstrate that specifically in the case of decomposition in the penultimate layer, it exists optimal methods for importance estimation, namely RISE [32],

Integrated Gradients [7], Gradient-Input [6], and Occlusion [13]. We provide theoretical evidence to support the optimality of these methods.

**From concept importance to attribution methods**    The dictionary learning formulation allows us to define the concepts $\mathbf{V}$ in such a way that they are optimal to reconstruct the activation, i.e., $\mathbf{A} \approx \mathbf{U}\mathbf{V}^\top$. Nevertheless, this does not guarantee that those concepts are important for the model's prediction. For example, the "grass" concept might be important to characterize the activations of a neural network when presented with a St-Bernard image, but it might not be crucial for the network to classify the same image as a St Bernard [26, 16, 54]. The notion of concept importance is precisely introduced to avoid such a confirmation bias and to identify the concepts used to classify among all detected concepts.

We use the notion of Concept ATtribution methods (which we denote as *CAT*s) to assess the concept importance score. The CATs are a generalization of the attribution methods: while attribution methods assess the sensitivity of the model output to a change in the pixel space, the concept importance evaluates the sensitivity to a change in the concept space. To compute the CATs methods, it is necessary to link the activation $\boldsymbol{a} \in \mathbb{R}^p$ to the concept base $\mathbf{V}$ and the model prediction $\boldsymbol{y}$. To do so, we feed the second part of the network ($\boldsymbol{g}$) with the activation reconstruction ($\boldsymbol{u}\mathbf{V}^\top \approx \boldsymbol{a}$) so that $\boldsymbol{y} = \boldsymbol{g}(\boldsymbol{u}\mathbf{V}^\top)$. Intuitively, a CAT method quantifies how a variation of $\boldsymbol{u}$ will impact $\boldsymbol{y}$. We denote $\varphi_i(\boldsymbol{u})$ the $i$-th coordinate of $\varphi(\boldsymbol{u})$, so that it represents the importance of the $i$-th concept in the representation $\boldsymbol{u}$. Equipped with these notations, we can leverage the sensitivity metrics introduced in standard attribution methods to re-define the current measures of concept importance, as well as introduce the new CATs borrowed from the attribution methods literature:

$$
\varphi_i(\boldsymbol{u}) = 
\begin{cases}
\nabla_{\boldsymbol{u}_i}\boldsymbol{g}(\boldsymbol{u}\mathbf{V}^\top) & \text{(used in TCAV: } \textbf{\textcolor{purple}{ACE}}, \textcolor{red}{\textbf{ICE}}\text{)}, \quad (5) \\[2mm]
\dfrac{\mathbb{E}_{\mathbf{m}_{\sim i}}(\mathbb{V}_{\mathbf{m}}(\boldsymbol{g}((\boldsymbol{u}\odot\mathbf{m})\mathbf{V}^\top)|\mathbf{m}_{\sim i}))}{\mathbb{V}(\boldsymbol{g}((\boldsymbol{u}\odot\mathbf{m})\mathbf{V}^\top))} & \text{(Sobol : } \textbf{\textcolor{purple}{CRAFT}}\text{)}, \quad (6) \\[2mm]
(\boldsymbol{u}_i - \boldsymbol{u}_i') \times \displaystyle\int_0^1 \nabla_{\boldsymbol{u}_i}\boldsymbol{g}((\boldsymbol{u}'\alpha + (1-\alpha)(\boldsymbol{u}-\boldsymbol{u}'))\mathbf{V}^\top)d\alpha & \text{Int.Gradients}, \quad (7) \\[2mm]
\mathbb{E}_{\boldsymbol{\delta}\sim\mathcal{N}(0,\mathbf{I}\sigma)}(\nabla_{\boldsymbol{u}_i}\boldsymbol{g}((\boldsymbol{u}+\boldsymbol{\delta})\mathbf{V}^T)) & \text{Smooth grad.} \quad (8) \\[2mm]
\dots
\end{cases}
$$

The complete derivation of the 7 new CATs is provided in the appendix. In the derivations, $\nabla_{\boldsymbol{u}_i}$ denotes the gradient with respect to the $i$-th coordinate of $\boldsymbol{u}$, while $\mathbb{E}$ and $\mathbb{V}$ represent the expectation and variance, respectively. In Eq. 6, $\mathbf{m}$ is a mask of real-valued random variable between 0 and 1 (i.e $\mathbf{m} \sim \mathcal{U}([0,1]^p)$). We note that, when we use the gradient (w.r.t to $\boldsymbol{u}_i$) as an importance score, we end up with the directional derivative used in the TCAV metric [26], which is used by **ACE** and **ICE** to assess the importance of concepts. **CRAFT** leverages the Sobol-Hoeffding decomposition (used in sensitivity analysis), to estimate the concept importance. The Sobol indices measure the contribution of a concept as well as its interaction of any order with any other concepts to the output variance. Intuitively, the numerator of Eq. 6 is the expected variance that would be left if all variables but $\boldsymbol{u}_i$ were to be fixed.

**Evaluation of concept importance methods**    Our generalization of the concept importance score, using the Concept ATtributions (CATs), allows us to observe that current concept-based explainability methods are only leveraging a small subset of concept importance methods. In Appendix A, we provide the complete derivation of 7 new CATs based on the following existing attribution methods, notably: Gradient input [6], Smooth grad [5], Integrated Gradients [7], VarGrad [55], Occlusion [13], HSIC [10] and RISE [32].

With the concept importance scoring now formulated as a generalization of attribution methods, we can borrow the metrics from the attribution domain to evaluate the faithfulness [56, 32, 33] of concept importance methods. In particular, we adapt three distinct metrics to evaluate the significance of concept importance scores: the C-Deletion [32], C-Insertion [32], and C-$\mu$Fidelity [33] metrics. In C-Deletion, we gradually remove the concepts (as shown in Figure 4), in decreasing order of importance, and we report the network's output each time a concept is removed. When a concept is removed in C-Deletion, the corresponding coordinate in the representation is set to $\mathbf{0}$. The final C-Deletion metrics are computed as the area under the curve in Figure 4. For C-Insertion, this is

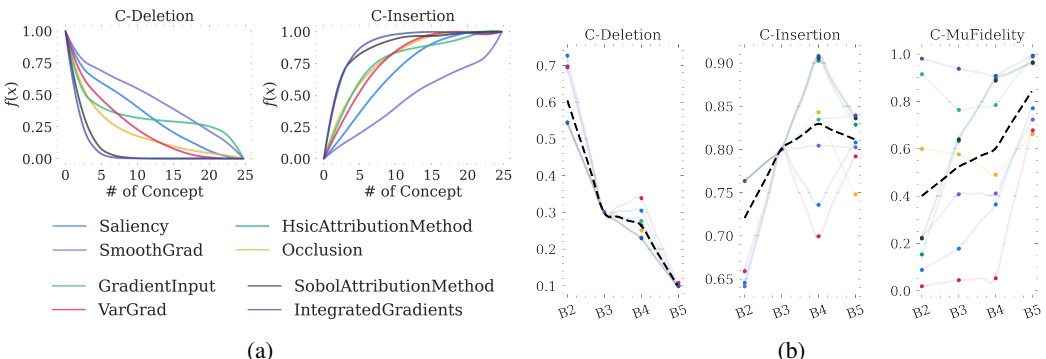

(a)                                                    (b)

Figure 4: **(a) C-Deletion, C-Insertion curves.** Fidelity curves for C-Deletion depict the model's score as the most important concepts are removed. The results are averaged across 10 classes of ImageNet using a ResNet50 model. **(b) C-Deletion, C-Insertion and C-$\mu$Fidelity across layer.** We report the 3 metrics to evaluate CATs for each block (from B2 to B5) of a ResNet50. We evaluate 8 Concept Attribution methods, all represented with different colors (see legend in Figure 4(a). The average trend of these eight methods is represented by the black dashed line (- - -). Lower C-Deletion is better, higher C-Insertion and C-$\mu$Fidelity is better. Overall, it appears that the estimation of importance becomes more faithful towards the end of the model.

the opposite: we start from a representation vector filled with zero, and we progressively add more concepts, following an increasing order of importance.

For the C-$\mu$Fidelity, we calculate the correlation between the model's output when concepts are randomly removed and the importance assigned to those specific concepts. The results across layers for a ResNet50 model are depicted in Figure 4b. We observe that decomposition towards the end of the model is preferred across all the metrics. As a result, in the next section, we will specifically examine the case of the penultimate layer.

**A note on the last layer**  Based on our empirical results, it appears that the last layer is preferable for both improved concept extraction and more accurate estimation of importance. Herein, we derive theoretical guarantees about the optimality of concept importance methods in the penultimate layer. Without loss of generality, we assume $y \in \mathbb{R}$ the logits of the class of interest. In the penultimate layer, the score $y$ is a linear combination of activations: $y = \boldsymbol{a}\mathbf{W} + \mathbf{b}$ for weight matrix $\mathbf{W}$ and bias $\mathbf{b}$. In this particular case, all CATs have a closed-form (see appendix B), that allows us to derive 2 theorems. The first theorem tackles the CATs optimality for the C-Deletion and C-Insertion methods (demonstration in Appendix D). We observe that the C-Deletion and C-Insertion problems can be represented as weighted matroids. Therefore the greedy algorithms lead to optimal solutions for CATs and a similar theorem could be derived for C-$\mu$Fidelity.

**Theorem 3.1** (Optimal C-Deletion, C-Insertion in the penultimate layer). *When decomposing in the penultimate layer, **Gradient Input**, **Integrated Gradients**, **Occlusion**, and **Rise** yield the optimal solution for the C-Deletion and C-Insertion metrics. More generally, any method $\boldsymbol{\varphi}(\boldsymbol{u})$ that satisfies the condition $\forall (i, j) \in \{1, \ldots, k\}^2, (\boldsymbol{u} \odot \mathbf{e}_i)\mathbf{V}^\mathsf{T}\mathbf{W} \geq (\boldsymbol{u} \odot \mathbf{e}_j)\mathbf{V}^\mathsf{T}\mathbf{W} \implies \boldsymbol{\varphi}(\boldsymbol{u})_i \geq \boldsymbol{\varphi}(\boldsymbol{u})_j$ yields the optimal solution.*

**Theorem 3.2** (Optimal C-$\mu$Fidelity in the penultimate layer). *When decomposing in the penultimate layer, **Gradient Input**, **Integrated Gradients**, **Occlusion**, and **Rise** yield the optimal solution for the C-$\mu$Fidelity metric.*

Therefore, for all 3 metrics, the concept importance methods based on Gradient Input, Integrated Gradient, Occlusion, and Rise are optimal, when used in the penultimate layer.

In summary, our investigation of concept extraction methods from the perspective of dictionary learning demonstrates that the NMF approach, specifically when extracting concepts from the penultimate layer, presents the most appealing trade-off compared to PCA and K-Means methods. In addition, our formalization of concept importance using attribution methods provided us with a theoretical guarantee for 4 different CATs. Henceforth, we will then consider the following setup: a

NMF on the penultimate layer to extract the concepts, combined with a concept importance method based on Integrated Gradient.

### 3.3 Unveiling main strategies

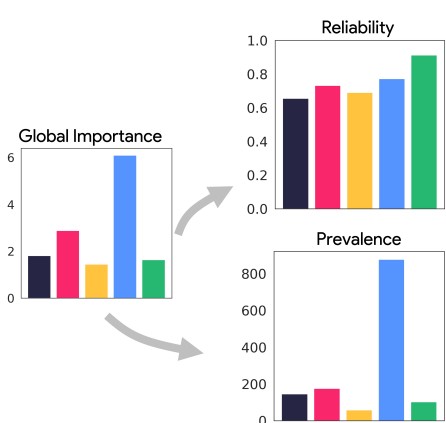

Figure 5: **From global (class-based) to local (image-based) importance.** Global importance can be decomposed into *reliability* and *prevalence* scores. Prevalence quantifies how frequently a concept is encountered, and reliability indicates how diagnostic a concept is for the class. The bar-charts are computed for the class "Espresso" on a ResNet50 (see Figure 1, left panel)

So far, the concept-based explainability methods have mainly focused on evaluating the global importance of concepts, i.e., the importance of concepts for an entire class [26, 29]. This point can be limiting when studying misclassified data points, as we can speculate that the most important concepts for a given class might not hold for an individual sample (local importance). Fortunately, our formulation of concept importance using attribution methods gives us access to importance scores at the level of individual samples (*i.e.,* $\varphi(\boldsymbol{u})$). Here, we show how to use these local importance scores to efficiently cluster data points based on the strategy used for their classification.

The local (or image-based) importance of concepts can be integrated into global measures of importance for the entire class with the notion of *prevalence* and *reliability* (see Figure 5). A concept is said to be prevalent at the class level when it appears very frequently. A *prevalence* score is computed based on the number of times a concept is identified as the most important one, i.e., $\arg\max\varphi(\boldsymbol{u})$. At the same time, a concept is said to be reliable if it is very likely to trigger a correct prediction. The *reliability* is quantified using the mean classification accuracy on samples sharing the same most important concept.

**Strategic cluster graph.** In the strategic cluster graph (Figure 1 and Figure 6), we combine the notions of concept *prevalence* and *reliability* to reveal the main strategies of a model for a given category, more precisely, we reveal their repartition across the different samples of the class. We use a dimensionality reduction technique (UMAP [57]) to arrange the data points based on the concept importance vector $\varphi(\boldsymbol{u})$ of each sample. Data points are colored according to the associated concept with the highest importance – $\arg\max\varphi(\boldsymbol{u})$. Interestingly, one can see in Figure 1 and Figure 6 that spatially close points represent samples classified using *similar strategies* – as they exhibit similar concept importance – and not necessarily similar embeddings. For example, for the "lemon" object category (Figure 6), the texture of the lemon peel is the most *prevalent* concept, as it appears to be the dominant concept in $90\%$ of the samples (see the green cluster in Figure 6). We also observe that the concept "pile of round, yellow objects" is not reliable for the network to properly classify a lemon as it results in a mean classification accuracy of $40\%$ only (see top-left graph in Figure 6).

In Figure 6 (right panel), we have exploited the strategic cluster graph to understand the classification strategies leading to bad classifications. For example, an orange ($1^{st}$ image, $1^{st}$ row) was classified as a lemon because of the peel texture they both share. Similarly, a cathedral roof was classified as a lemon because of the wedge-shaped structure of the structure ($4^{th}$ image, $1^{st}$ row).

## 4 Discussion

This article introduced a theoretical framework that unifies all modern concept-based explainability methods. Breaking down and formalizing the two essential steps in these methods, concept extraction and concept importance scoring, allowed us to better understand the underlying principles driving concept-based explainability. We leveraged this unified framework to propose new evaluation metrics for assessing the quality of extracted concepts. Through experimental and theoretical analyses, we justified the standard use of the last layer of an ANN for concept-based explanation. Finally, we harnessed the parallel between concept importance and attribution methods to gain insights into global concept importance (at the class level) by examining local concept importance (for individual samples). We proposed the strategic cluster graph, which provides insights into the strategy used by an ANN to classify images. We have provided an example use of this approach to better understand

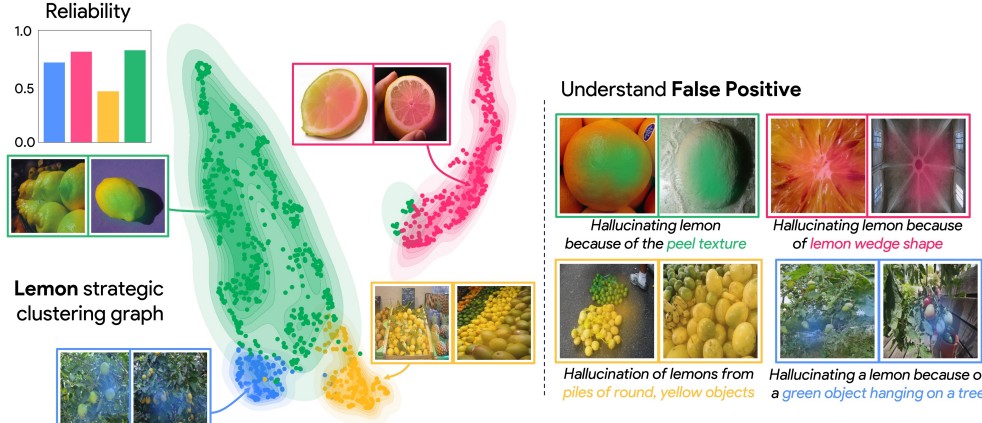

Figure 6: **Strategic cluster graph for the lemon category. Left**: U-MAP of lemon samples, in the concept space. Each concept is represented with its own color and is exemplified with example belonging to the cluster. The concepts are ● the lemon wedge shape, ● a pile of round, yellow objects, ● green objects hanging on a tree, and finally ● the peel texture, which is the predominant strategy. The reliability of each concept is shown in the top-left bar-chart. **Right**: Example of images predicted as lemon along with their corresponding explanations. These misclassified images are recognized as lemons through the implementation of strategies that are captured by our proposed strategic cluster graph.

the failure cases of a system. Overall, our work demonstrates the potential benefits of the dictionary learning framework for automatic concept extraction and we hope this work will pave the way for further advancements in the field of XAI.

## Acknowledgements

This work was conducted as part of the DEEL project[†]. Funding was provided by ONR (N00014-19-1-2029), NSF (IIS-1912280 and EAR-1925481), DARPA (D19AC00015), NIH/NINDS (R21 NS 112743), and the ANR-3IA Artificial and Natural Intelligence Toulouse Institute (ANR-19-PI3A-0004). Additional support provided by the Carney Institute for Brain Science and the Center for Computation and Visualization (CCV). We acknowledge the Cloud TPU hardware resources that Google made available via the TensorFlow Research Cloud (TFRC) program as well as computing hardware supported by NIH Office of the Director grant S10OD025181.

---

[†]https://www.deel.ai/

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
