# A Attribution methods for Concepts

In the following section, we will re-derive the different attribution methods in the literature. We use the Xplique library and adapted each methods [58]. We quickly recall that we seek to estimate the importance of each concept for a set of concept coefficients $\boldsymbol{u} = (\boldsymbol{u}_1, \ldots, \boldsymbol{u}_k) \in \mathbb{R}^k$ in the concept basis $\mathbf{V} \in \mathbb{R}^{p \times k}$. This concept basis is a re-interpretation of a latent space $\mathcal{H} \subseteq \mathbb{R}^p$ and the function $g : \mathbb{R}^p \to \mathbb{R}$ is a signal used to compute importance from (e.g., logits value, cosine similarity with a sentence...). Each Attributions method will map a set of concept values to an importance score $\boldsymbol{\varphi} : \mathbb{R}^k \to \mathbb{R}^k$, a greater score $\boldsymbol{\varphi}(\boldsymbol{u})_i$ indicates that a concept $\boldsymbol{u}_i$ is more important.

**Saliency (SA)** [4] was originally a visualization technique based on the gradient of a class score relative to the input, indicating in an infinitesimal neighborhood, which pixels must be modified to most affect the score of the class of interest. In our case, it indicates which concept in an infinitesimal neighborhood has the most influence on the output:

$$\boldsymbol{\varphi}^{(SA)}(\boldsymbol{u}) = \nabla_{\boldsymbol{u}} \boldsymbol{g}(\boldsymbol{u} V^{\mathsf{T}}).$$

**Gradient $\odot$ Input (GI)** [6] is based on the gradient of a class score relative to the input, element-wise with the input, it was introduced to improve the sharpness of the attribution maps. A theoretical analysis conducted by [59] showed that Gradient $\odot$ Input is equivalent to $\epsilon$-LRP and DeepLIFT [6] methods under certain conditions – using a baseline of zero, and with all biases to zero. In our case, it boils down to:

$$\boldsymbol{\varphi}^{(GI)}(\boldsymbol{u}) = \boldsymbol{u} \odot \nabla_{\boldsymbol{u}} \boldsymbol{g}(\boldsymbol{u} \mathbf{V}^{\mathsf{T}}).$$

**Integrated Gradients (IG)** [7] consists of summing the gradient values along the path from a baseline state to the current value. The baseline $\boldsymbol{u}_0$ used is zero. This integral can be approximated with a set of $m$ points at regular intervals between the baseline and the point of interest. In order to approximate from a finite number of steps, we use a trapezoidal rule and not a left-Riemann summation, which allows for more accurate results and improved performance (see [60] for a comparison). For all the experiments $m = 30$.

$$\boldsymbol{\varphi}^{(IG)}(\boldsymbol{u}) = (\boldsymbol{u} - \boldsymbol{u}_0) \int_0^1 \nabla_{\boldsymbol{u}} \boldsymbol{g}((\boldsymbol{u}_0 + \alpha(\boldsymbol{u} - \boldsymbol{u}_0))\mathbf{V}^{\mathsf{T}}) \, \mathrm{d}\alpha.$$

**SmoothGrad (SG)** [5] is also a gradient-based explanation method, which, as the name suggests, averages the gradient at several points corresponding to small perturbations (drawn i.i.d from an isotropic normal distribution of standard deviation $\sigma$) around the point of interest. The smoothing effect induced by the average helps to reduce the visual noise, and hence improves the explanations. In our case, the attribution is obtained after averaging $m$ points with noise added to the concept coefficients. For all the experiments, we took $m = 30$ and $\sigma = 0.1$.

$$\boldsymbol{\varphi}^{(SG)}(\boldsymbol{u}) = \mathbb{E}_{\boldsymbol{\delta} \sim \mathcal{N}(0, \mathbf{I}\sigma)} (\nabla_{\boldsymbol{u}} \boldsymbol{g}(\boldsymbol{u} + \boldsymbol{\delta})).$$

**VarGrad (VG)** [55] was proposed as an alternative to SmoothGrad as it employs the same methodology to construct the attribution maps: using a set of $m$ noisy inputs, it aggregates the gradients using the variance rather than the mean. For the experiment, $m$ and $\sigma$ are the same as SmoothGrad. Formally:

$$\boldsymbol{\varphi}^{(VG)}(\boldsymbol{u}) = \mathbb{V}_{\boldsymbol{\delta} \sim \mathcal{N}(0, \mathbf{I}\sigma)} (\nabla_{\boldsymbol{u}} \boldsymbol{g}(\boldsymbol{u} + \boldsymbol{\delta})).$$

**Occlusion (OC)** [61] is a simple – yet effective – sensitivity method that sweeps a patch that occludes pixels over the images using a baseline state and use the variations of the model prediction to deduce critical areas. In our case, we simply omit each concept one-at-a-time to deduce the concept's importance. For all the experiments, the baseline state $\boldsymbol{u}_0$ was zero.

$$\boldsymbol{\varphi}^{(OC)}(\boldsymbol{u})_i = \boldsymbol{g}(\boldsymbol{u} \mathbf{V}^{\mathsf{T}}) - \boldsymbol{g}(\boldsymbol{u}_{[i = \boldsymbol{u}_0]} \mathbf{V}^{\mathsf{T}})$$

**Sobol Attribution Method (SM)** [9] then used for estimating concept importance in [29] is a black-box attribution method grounded in Sensitivity Analysis. Beyond modeling the individual contributions of image regions, Sobol indices provide an efficient way to capture higher-order interactions between image regions and their contributions to a neural network's prediction through the lens of variance. In our case, the score for a concept $\boldsymbol{u}_i$ is the expected variance that would be left if all variables but $i$ were to be fixed :

$$\varphi^{(SM)}(\boldsymbol{u})_i = \frac{\mathbb{E}(\mathbb{V}(\boldsymbol{g}((\boldsymbol{u} \odot \mathbf{M})\mathbf{V}^\mathsf{T})|\mathbf{M}_{\sim i}))}{\mathbb{V}(\boldsymbol{g}((\boldsymbol{u} \odot \mathbf{M})\mathbf{V}^\mathsf{T}))}.$$

With $\mathbf{M} \sim \mathcal{U}([0,1])^k$. For all the experiments, the number of designs was 32 and we use the Jansen estimator of the Xplique library.

**HSIC Attribution Method (HS)** [10] seeks to explain a neural network's prediction for a given input image by assessing the dependence between the output and patches of the input. In our case, we randomly mask/remove concepts and measure the dependence between the output and the presence of each concept through $N$ binary masks. Formally:

$$\varphi^{(HS)}(\boldsymbol{u}) = \frac{1}{(N-1)^2}\mathrm{Tr}(KHLH).$$

With $H, L, K \in \mathbb{R}^{N \times N}$ and $K_{ij} = k(\mathbf{M}_i, \mathbf{M}_j)$, $L_{ij} = l(\boldsymbol{y}_i, \boldsymbol{y}_j)$ and $H_{ij} = \delta(i = j) - N^{-1}$. Here, $k(\cdot, \cdot)$ and $l(\cdot, \cdot)$ denote the chosen kernels and $\mathbf{M} \sim \{0, 1\}^p$ the binary mask applied to the input $\boldsymbol{u}$.

**RISE (RI)** [32] is also a black-box attribution method that probes the model with multiple version of a masked input to model the most important features. Formally, with $\boldsymbol{m} \sim \mathcal{U}([0,1])^k$. :

$$\varphi_i^{(RI)}(\boldsymbol{u}) = \mathbb{E}(\boldsymbol{g}(\boldsymbol{u} \odot \boldsymbol{m})|\boldsymbol{m}_i = 1).$$

# B   Closed-form of Attributions for the last layer

Without loss of generality, we focus on the decomposition in the last layer, that is $\boldsymbol{a} = \boldsymbol{u}\mathbf{V}^\mathsf{T}$ with parameters $(\mathbf{W}, \mathbf{b})$ for the weight and the bias respectively, hence we obtain $\boldsymbol{y} = (\boldsymbol{u}\mathbf{V}^\mathsf{T})\mathbf{W} + \mathbf{b}$ with $\mathbf{W} \in \mathbb{R}^p$ and $\mathbf{b} \in \mathbb{R}$.

We start by deriving the closed form of Saliency (SA) and naturally Gradient-Input (GI):

$$\begin{aligned}
\varphi^{(SA)}(\boldsymbol{u}) &= \nabla_{\boldsymbol{u}}\boldsymbol{g}(\boldsymbol{u}\mathbf{V}^\mathsf{T}) = \nabla_{\boldsymbol{u}}(\boldsymbol{u}\mathbf{V}^\mathsf{T}\mathbf{W} + \mathbf{b}) \\
&= \mathbf{W}^\mathsf{T}\mathbf{V}
\end{aligned}$$

$$\begin{aligned}
\varphi^{(GI)}(\boldsymbol{u}) &= \nabla_{\boldsymbol{u}}\boldsymbol{g}(\boldsymbol{u}\mathbf{V}^\mathsf{T}) \odot \boldsymbol{u} = \nabla_{\boldsymbol{u}}(\boldsymbol{u}\mathbf{V}^\mathsf{T}\mathbf{W} + \mathbf{b}) \odot \boldsymbol{u} \\
&= \mathbf{W}^\mathsf{T}\mathbf{V} \odot \boldsymbol{u}
\end{aligned}$$

We observe two different forms that will in fact be repeated for the other methods, for example with Integrated-Gradient (IG) which will take the form of Gradient-Input, while SmoothGrad (SG) will take the form of Saliency.

$$\begin{aligned}
\varphi^{(IG)}(\boldsymbol{u}) &= (\boldsymbol{u} - \boldsymbol{u}_0) \odot \int_0^1 \nabla_{\boldsymbol{u}}\boldsymbol{g}((\boldsymbol{u}_0 + \alpha(\boldsymbol{u} - \boldsymbol{u}_0))\mathbf{V}^\mathsf{T})\,\mathrm{d}\alpha \\
&= \boldsymbol{u} \odot \int_0^1 \nabla_{\boldsymbol{u}}((\alpha\boldsymbol{u}))\mathbf{V}^\mathsf{T}\mathbf{W} + \mathbf{b} + (\alpha - 1)\boldsymbol{u}_0\mathbf{V}^\mathsf{T}\mathbf{W})\,\mathrm{d}\alpha \\
&= \boldsymbol{u} \odot \int_0^1 \alpha\mathbf{W}^\mathsf{T}\,\mathrm{d}\alpha = \boldsymbol{u} \odot \mathbf{W}^\mathsf{T}\mathbf{V}\left[\frac{1}{2}\alpha^2\right]_0^1 \\
&= \frac{1}{2}\boldsymbol{u} \odot \mathbf{W}^\mathsf{T}\mathbf{V}.
\end{aligned}$$

$$\begin{aligned}
\varphi^{(SG)}(\boldsymbol{u}) &= \mathop{\mathbb{E}}_{\boldsymbol{\delta} \sim \mathcal{N}(0, \mathbf{I}\sigma)}(\nabla_{\boldsymbol{u}}\boldsymbol{g}(\boldsymbol{u} + \boldsymbol{\delta})) = \mathop{\mathbb{E}}_{\boldsymbol{\delta} \sim \mathcal{N}(0, \mathbf{I}\sigma)}(\nabla_{\boldsymbol{u}}((\boldsymbol{u} + \boldsymbol{\delta})\mathbf{V}^\mathsf{T}\mathbf{W} + \mathbf{b})) \\
&= \mathop{\mathbb{E}}_{\boldsymbol{\delta} \sim \mathcal{N}(0, \mathbf{I}\sigma)}(\nabla_{\boldsymbol{u}}(\boldsymbol{u}\mathbf{V}^\mathsf{T}\mathbf{W})) \\
&= \mathbf{W}^\mathsf{T}\mathbf{V}
\end{aligned}$$

The case of VarGrad is specific, as the gradient of a linear system being constant, its variance is null.

$$\varphi^{(VG)}(\boldsymbol{u}) = \underset{\boldsymbol{\delta} \sim \mathcal{N}(0,\mathbf{I}\sigma)}{\mathbb{V}} (\nabla_{\boldsymbol{u}} g(\boldsymbol{u} + \boldsymbol{\delta})) = \underset{\boldsymbol{\delta} \sim \mathcal{N}(0,\mathbf{I}\sigma)}{\mathbb{V}} (\nabla_{\boldsymbol{u}}((\boldsymbol{u} + \boldsymbol{\delta})\mathbf{V}^{\mathsf{T}}\mathbf{W} + \mathbf{b}))$$

$$= \underset{\boldsymbol{\delta} \sim \mathcal{N}(0,\mathbf{I}\sigma)}{\mathbb{V}} (\mathbf{W}^{\mathsf{T}}\mathbf{V})$$

$$= 0$$

Finally, for Occlusion (OC) and RISE (RI), we fall back on the Gradient Input form (with multiplicative and additive constant for RISE).

$$\varphi_i^{(OC)}(\boldsymbol{u}) = g(\boldsymbol{u}\mathbf{V}^{\mathsf{T}}) - g(\boldsymbol{u}_{[i=\boldsymbol{u}_0]}\mathbf{V}^{\mathsf{T}}) = \boldsymbol{u}\mathbf{V}^{\mathsf{T}}\mathbf{W} + \mathbf{b} - (\boldsymbol{u}_{[i=\boldsymbol{u}_0]}\mathbf{V}^{\mathsf{T}}\mathbf{W} + \mathbf{b})$$

$$= (\sum_j^r \boldsymbol{u}_j \mathbf{V}_j^{\mathsf{T}})\mathbf{W} - (\sum_{j \neq i}^r \boldsymbol{u}_j \mathbf{V}_j^{\mathsf{T}})\mathbf{W}$$

$$= \boldsymbol{u}_i \mathbf{V}_i^{\mathsf{T}}\mathbf{W}$$

thus $\varphi^{(OC)}(\boldsymbol{u}) = \boldsymbol{u} \odot \mathbf{W}^{\mathsf{T}}\mathbf{V}$

$$\varphi_i^{(RI)}(\boldsymbol{u}) = \mathbb{E}(g(\boldsymbol{u} \odot \boldsymbol{m})|\boldsymbol{m}_i = 1) = \mathbb{E}((\boldsymbol{u} \odot \boldsymbol{m})\mathbf{V}^{\mathsf{T}}\mathbf{W} + \mathbf{b}|\boldsymbol{m}_i = 1)$$

$$= \mathbf{b} + \sum_{j \neq i}^r \boldsymbol{u}_j \mathbb{E}(\boldsymbol{m}_j)\mathbf{V}_j^{\mathsf{T}}\mathbf{W} + \boldsymbol{u}_i \mathbf{V}_i^{\mathsf{T}}\mathbf{W}$$

$$= \mathbf{b} + \frac{1}{2}(\boldsymbol{u}\mathbf{V}^{\mathsf{T}}\mathbf{W} + \boldsymbol{u}_i \mathbf{V}_i^{\mathsf{T}}\mathbf{W})$$

# C $\mu$ Fidelity optimality

Before showing that some methods are optimal with regard to C-Deletion and C-Insertion, we start with a first metric that studies the fidelity of the importance of concepts: $\mu$Fidelity, whose definition we recall

$$\mu F = \underset{\substack{S \subseteq \{1,\dots,k\} \\ |S|=m}}{\rho} (\sum_{i \in S} \varphi(\boldsymbol{u})_i, g(\boldsymbol{u}) - g(\boldsymbol{u}_{[\boldsymbol{u}_i=\boldsymbol{u}_0,i \in S]}))$$

With $\rho$ the Pearson correlation and $\boldsymbol{u}_{[\boldsymbol{u}_i=\boldsymbol{u}_0,i \in S]}$ means that all $i$ components of $\boldsymbol{u}$ are set to zero.

**Theorem C.1** (Optimal $\mu$Fidelity in the last layer). *When decomposing in the last layer, **Gradient Input**, **Integrated Gradients**, **Occlusion**, and **Rise** yield the optimal solution for the $\mu$Fidelity metric. In a more general sense, any method $\varphi(\boldsymbol{u})$ that is of the form $\varphi_i(\boldsymbol{u}) = a(\boldsymbol{u}_i \mathbf{V}_i^{\mathsf{T}}\mathbf{W}) + b$ with $a \in \mathbb{R}^+, b \in \mathbb{R}$ yield the optimal solution, thus having a correlation of 1.*

*Proof.* In the last layer case, $\mu$Fidelity boils down to:

$$\mu F = \underset{\substack{S \subseteq \{1,\dots,k\} \\ |S|=m}}{\rho} (\sum_{i \in S} \varphi(\boldsymbol{u})_i, \boldsymbol{u}\mathbf{V}^{\mathsf{T}}\mathbf{W} + \mathbf{b} - (\sum_{i \notin S} \boldsymbol{u}_i \mathbf{V}_i^{\mathsf{T}}\mathbf{W}) - \mathbf{b})$$

$$= \underset{\substack{S \subseteq \{1,\dots,k\} \\ |S|=m}}{\rho} (\sum_{i \in S} \varphi(\boldsymbol{u})_i, \sum_{i \in S} \boldsymbol{u}_i \mathbf{V}_i^{\mathsf{T}}\mathbf{W})$$

We recall that for **Gradient Input**, **Integrated Gradients**, **Occlusion**, $\varphi_i(\boldsymbol{u}) \propto \boldsymbol{u}_i \mathbf{V}_i^{\mathsf{T}}\mathbf{W}$, thus

$$\mu F = \underset{\substack{S \subseteq \{1,\dots,k\} \\ |S|=m}}{\rho} (\sum_{i \in S} \boldsymbol{u}_i \mathbf{V}_i^{\mathsf{T}}\mathbf{W}, \sum_{i \in S} \boldsymbol{u}_i \mathbf{V}_i^{\mathsf{T}}\mathbf{W}) = 1$$

For **RISE**, we get the following characterization:

$$\mu F = \rho_{\substack{S \subseteq \{1,\ldots,k\} \\ |S|=m}} \Big( \sum_{i \in S} \mathbf{b} + \frac{1}{2}(\boldsymbol{u}\mathbf{V}^\mathsf{T}\mathbf{W} + \boldsymbol{u}_i\mathbf{V}_i^\mathsf{T}\mathbf{W}), \sum_{i \in S} \boldsymbol{u}_i\mathbf{V}_i^\mathsf{T}\mathbf{W} \Big)$$

$$= \rho_{\substack{S \subseteq \{1,\ldots,k\} \\ |S|=m}} \Big( |S|(\mathbf{b} + \frac{1}{2}(\boldsymbol{u}\mathbf{V}^\mathsf{T}\mathbf{W})) + \sum_{i \in S} \frac{1}{2}\boldsymbol{u}_i\mathbf{V}_i^\mathsf{T}\mathbf{W}, \sum_{i \in S} \boldsymbol{u}_i\mathbf{V}_i^\mathsf{T}\mathbf{W} \Big)$$

$$= \rho_{\substack{S \subseteq \{1,\ldots,k\} \\ |S|=m}} \Big( a(\sum_{i \in S} \boldsymbol{u}_i\mathbf{V}_i^\mathsf{T}\mathbf{W}) + b, \sum_{i \in S} \boldsymbol{u}_i\mathbf{V}_i^\mathsf{T}\mathbf{W} \Big) = 1$$

with $a = \frac{1}{2}, b = m(\mathbf{b} + \frac{1}{2}(\boldsymbol{u}\mathbf{V}^\mathsf{T}\mathbf{W}))$.

$\square$

# D Optimality for C-Insertion and C-Deletion

In order to prove the optimality of some attribution methods on the C-Insertion and C-Deletion metrics, we will use the Matroid theory of which we recall some fundamentals.

Matroids were introduced by Whitney in 1935 [62]. It was quickly realized that they unified properties of various domains such as graph theory, linear algebra or geometry. Later, in the '60s, a connection was made with combinatorial optimization, nothing that they also played a central role in combinatorial optimization.

The power of this tool is that it allows us to show easily that greedy algorithms are optimal with respect to some criterion on a broad range of problems. Here, we show that insertion is a greedy algorithm (since the concepts inserted are chosen sequentially based on the model score).

For the rest of this section, we assume $E = \{e_1, \ldots, e_k\}$ the set of the canonical vectors in $\mathbb{R}^k$, with $e_i$ being the element associated with the $i^{th}$ concept.

**Definition D.1** (Matroid). *A matroid $M$ is a tuple $(E, \mathcal{J})$, where $E$ is a finite ground set and $\mathcal{J} \subseteq 2^E$ is the power set of $E$, a collection of independent sets, such that:*

1. *$\mathcal{J}$ is nonempty, $\emptyset \in \mathcal{J}$.*

2. *$\mathcal{J}$ is downward closed; i.e., if $S \in \mathcal{J}$ and $S' \subseteq S$, then $S' \in \mathcal{J}$*

3. *If $S, S' \in \mathcal{J}^2$ and $|S| < |S'|$, then $\exists s \in S' \setminus S$ such that $S \cup \{s\} \in \mathcal{J}$*

In particular, we will need uniform matroids:

**Definition D.2** (Uniform Matroid). *Let $E$ be a set of size $k$ and let $n \in \{1, \ldots, k\}$. If $\mathcal{J}$ is the collection of all subsets of $E$ of size at most $n$, then $(E, \mathcal{J})$ is a matroid, called a uniform matroid and denoted $M^{(n)}$.*

Finally, we need to characterize the concept set chosen at each step.

**Definition D.3** (Base of Matroid). *Let $M = (E, \mathcal{J})$ be a matroid. A subset $B$ of $E$ is called a basis of $M$ if and only if:*

1. *$B \in \mathcal{J}$*

2. *$\forall e \in E \setminus B, \ B \cup \{e\} \notin \mathcal{J}$*

*Moreover, we denote $\mathcal{B}(M)$ the set of all the basis of $M$.*

At each step, the insertion metric selects the concepts of maximum score given a cardinality constraint. At each new step, the concepts from the previous step are selected and it add a new concept from the whole available set, the one not selected so far with the highest score. This criterion requires an additional ingredient: the *weight* associated to each element of the matroid - here an element of the matroid is a concept.

**Ponderated Matroid** Let $M^{(n)} = (E, \mathcal{J})$ be a uniform matroid and $w : E \to \mathbb{R}$ a weighting function associated to an element of $E$ (a concept). The goal of C-Insertion at step $n$ is to find a basis (a set of concepts) $B^\star$ subject to $|B| = n$, that maximizes the weighting function :

$$\forall B \in \mathcal{J}, \quad \sum_{e \in B^\star} w(e) \geq \sum_{e \in B} w(e).$$

Such a basis is called the basis of maximum weights (MW) of the weighted matroid $M^{(n)}$. We will see that the greedy algorithm associated with this weighting function gives the optimal solution to the MW problem on C-Insertion. First, let's define the *Greedy algorithm*.

---

**Algorithm 1** Greedy algorithm

---

**Require:** A $n$-uniform weighted matroid $M^{(n)} = (E, \mathcal{J}, w)$
  Sort the concepts by their weight $w(e_i)$ in non-increasing order, and store them in a list $\bar{e}$ such that $\forall (i, j) \subseteq \{1, \dots, k\}^2, w(\bar{e}_i) \geq w(\bar{e}_j)$ if $i < j$.
  $B^\star = \{\}$
  **for** $k = 1$ to $n$ **do**
    $B^\star = B^\star \cup \bar{e}_i$
  **end for**
  **return** $B^\star$

---

**Theorem D.4** (Greedy Algorithm is an optimal solution to MW.). *Let $M = (E, \mathcal{J}, w)$ a weighted matroid. The greedy Algorithm 1 returns a maximum basis of $M$.*

*Proof.* First, by definition, $B^\star$ is a basis and thus an independent set, i.e., $B^\star \in \mathcal{B}(M)$ (as $\forall (e, e') \in E^2$, $\langle e, e' \rangle = 0$). Now, suppose by contradiction that there exists a base $B'$ with a weight strictly greater than $B^\star$. We will obtain a contradiction with respect to the augmentation axiom of the matroid definition. Let $e_1, \dots, e_k$ be the elements of $M$ sorted such that $w(e_i) > w(e_j)$ whenever $i < j$. Let $n$ be the rank of our weighted uniform matroid $M^{(n)}$. Then we can write $B^\star = (e_{i_1}, \dots, e_{i_n})$ and $B' = (e_{j_1}, \dots, e_{j_n})$ with $j_k < j_l$ and $i_k < i_l$ for any $k < l$.

Let $\ell$ be the smallest positive integer such that $i_\ell > j_\ell$. In particular, $\ell$ exists and is at most $n$ by assumption. Consider the independent set $S_{\ell-1} = \{e_{i_1}, \dots e_{\ell-1}\}$ (in particular, $S_{\ell-1} = \emptyset$ if $\ell = 1$). According to the augmentation axiom (Definition D.2, I3), there exist $k \in \{1, \dots, \ell\}$ such that $S_{\ell-1} + e_{j_k} \in \mathcal{J}$ and $e_{j_k} \notin S_{\ell-1}$. However, $j_k \leq j_\ell < i_\ell$, thus $w(e_{j_k}) \leq w(e_{j_\ell}) < w(e_{i_\ell})$. This contradicts the definition of the greedy algorithm. $\qquad\square$

Now, we notice that for the last layer, Insertion is a weighted matroid. We insist that this result is *only true for the concepts in the penultimate layer*, as our demonstrations rely on the linearity of the decomposition. Here, the weight is given by the score of the model, which is a linear combination of concepts.

**Theorem D.5** (Optimal Insertion in the last layer). *When decomposing in the last layer, **Gradient Input**, **Integrated Gradients**, **Occlusion**, and **Rise** yield the optimal solution for the C-Insertion metric. In a more general sense, any method $\varphi(\boldsymbol{u})$ that satisfies the condition $\forall (i, j) \in \{1, \dots, k\}^2, (\boldsymbol{u} \odot \mathbf{e}_i)\mathbf{V}^\top \mathbf{W} \geq (\boldsymbol{u} \odot \mathbf{e}_j)\mathbf{V}^\top \mathbf{W} \implies \varphi(\boldsymbol{u})_i \geq \varphi(\boldsymbol{u})_j$ yield the optimal solution.*

*Proof.* Each $n$ step of the C-Insertion algorithm corresponds to the $n$-uniform weighted matroid with weighting function $w(e_i) = (\boldsymbol{u} \odot e_i)\mathbf{V}^\top \mathbf{W} + b = \boldsymbol{u}_i \mathbf{V}^\top \mathbf{W} + b$. Therefore, any $\varphi(\cdot)$ method that produces the same ordering as $w(\cdot)$ will yield the optimal solution. It easily follows that **Gradient Input**, **Integrated Gradients**, **Occlusion** are optimal as they all boil down to $\varphi_i(\boldsymbol{u}) = \boldsymbol{u}_i \mathbf{V}^\top \mathbf{W} + b$. Concerning RISE, suppose that $w(e_i) \geq w(e_j)$, then $\boldsymbol{u}_i \mathbf{V}_i^\top \mathbf{W} + b \geq \boldsymbol{u}_j \mathbf{V}_j^\top \mathbf{W} + b$, and $\varphi_i^{(RI)}(\boldsymbol{u}) - \varphi_j^{(RI)}(\boldsymbol{u}) = \mathbf{b} + \frac{1}{2}(\boldsymbol{u}\mathbf{V}^\top\mathbf{W} + \boldsymbol{u}_i\mathbf{V}_i^\top\mathbf{W}) - \mathbf{b} + \frac{1}{2}(\boldsymbol{u}\mathbf{V}^\top\mathbf{W} + \boldsymbol{u}_j\mathbf{V}_j^\top\mathbf{W}) = \boldsymbol{u}_i\mathbf{V}_i^\top\mathbf{W} - \boldsymbol{u}_j\mathbf{V}_j^\top\mathbf{W} \geq 0$. Thus, RISE importance will order in the same manner and is also optimal. $\qquad\square$

**Corollary D.6** (Optimal Deletion in the last layer). *When decomposing in the last layer, **Gradient Input**, **Integrated Gradients**, **Occlusion**, and **Rise** yield the optimal solution for the C-Deletion metric.*

*Proof.* It is simply observed that the C-Deletion problem seeks a minimum weight basis and corresponds to the same weighted matroid with weighting function $w'(\cdot) = -w(\cdot)$. $\qquad\square$

# E   Sparse Autoencoder

As a remainder, a general method (as it encompasses both PCA and K-means) to obtain the loading-dictionary pair and achieve a matrix reconstruction $\mathbf{A} = \mathbf{U}\mathbf{V}^\top$ is to train a neural network to obtain $\mathbf{U}$ from $\mathbf{A}$ such that the reconstruction of $\mathbf{A}$ is linear in $\mathbf{U}$. This can be formally represented as:

$$(\mathbf{U}^\star, \mathbf{V}^\star) = \arg\min_{\boldsymbol{\psi}, \mathbf{V}} \|\mathbf{A} - \boldsymbol{\psi}(\mathbf{A})\mathbf{V}^\top\|_F^2$$

Here, $\mathbf{U} = \psi(\mathbf{A})$. An interesting characteristic of NMF and K-means is the non-linear relationship between $\mathbf{A}$ and $\mathbf{U}$. Specifically, the transformation from $\mathbf{A}$ to $\mathbf{U}$ is non-linear, while the transformation from $\mathbf{U}$ to $\mathbf{A}$ is linear, as explained in [58], which need to introduce a method based on implicit differentiation to obtain the gradient of $\mathbf{U}$ with respect to $\mathbf{A}$. Indeed, the sequence of operations to optimize $\mathbf{U}$ causes us to lose information about which elements of $\mathbf{A}$ contributed to obtaining $\mathbf{U}$. We believe that this non-linear relationship (absent in PCA) may be an essential ingredient for effective concept extraction.

Finally, as described in this article, other characteristics that appear to make it interpretable include its compositionality (due to non-extreme sparsity), good reconstruction, and positivity, which aids in interpretation. Thus, the architecture of $\psi$ used for Figure 2 consists of a sequence of dense layers and batch normalization with ReLU activation to obtain positive scores and sparsity similar to NMF, without imposing constraints on $\mathbf{V}$. More formally, $\psi$ is a sequence of layers as follows:

$$\text{DENSE(128) - BATCHNORMALIZATION - RELU}$$
$$\text{DENSE(64) - BATCHNORMALIZATION - RELU}$$
$$\text{DENSE(10) - BATCHNORMALIZATION - RELU}$$

While the vector $\mathbf{V}$ is initialized using a truncated SVD [63]. We used Adam optimizer[64] with a learning rate of $1e^{-3}$. However, it's worth noting that there is a wealth of literature on dictionary learning that remains to be explored for the task of concept extraction [65].