# OpenReview forum: "A Holistic Approach to Unifying Automatic Concept Extraction and Concept Importance Estimation"
_NeurIPS.cc/2023/Conference — NeurIPS 2023 spotlight_

### Official Review · Reviewer_hQqh · 2023-07-04

**Soundness:** 2 fair
**Presentation:** 3 good
**Contribution:** 2 fair
**Rating:** 5
**Confidence:** 4

**Summary:**

This paper proposes a unifying framework to study common concept-based explainability methods. In particular, the authors observe that all the methods boil down to dictionary learning in the latent spaces of a neural network. Based on this observation, they (1) compare the different concept extraction/discovery methods and (2) extend established feature attribution methods to concept-based explanations. The authors investigation leads them to conclude that the penultimate layer is the best one where to extract concepts proving the optimality of concept-adapted metrics for attribution.

**Strengths:**

- (Clarity) Paper is in general easy to follow and fairly clear at discussing the core concepts.
- (Novelty) I am personally a big fan of unifying frameworks, so this paper seems like a reasonable steps towards understanding concept-based explainability methods
- (Significance) The strategic cluster graph is quite compelling from an user-experience perspective
- (Significance) While the results don't seem particularly surprising to me (more below), it is nice to see an analysis of concept-based methods laid out.
- (Quality) Mathematical derivations seem sound.

**Weaknesses:**

1) *Novelty/Significance*
  - *Theoretical results*: As mentioned above, I do not find the theoretical results particularly surprising. More details are reported below, but, in a nutshell, the results seem to me like a trivial extension of the work from Ancona et al. [1].
  - *Strategic cluster graph*: I would have preferred to see a more in-depth/quantitative analysis of this "tool". The concept analysis of the graph is fairly hand-wavy and arbitrary, e.g. in the naming of the concepts (apart for one of the wrongly classified samples).

As a more actionable feedback, perhaps I would comment that personally I would have preferred to have a less extensive theoretical analysis (since I find it unsurprising), and a more detailed use case for the strategic cluster graph, maybe with an user study. Or alternatively the authors could have leveraged their framework to propose a new method for concept extraction inspired by other dictionary learning method, perhaps something probabilistic such as the latent dirichlet allocation (although on the top of my mind, it may not be very trivial).

2) *Clarity*: some definitions are left unreferenced. For example (L. 273), there is no reference to the original feature-based metrics, so it might appear unclear how the C-metrics are an extension of those ones.

3) More detailed feedback can be found below, in the "Questions" section.

[1] Ancona, Marco, et al. "Towards better understanding of gradient-based attribution methods for deep neural networks." arXiv preprint arXiv:1711.06104 (2017).

**Questions:**

1. As mentioned above, using the authors framework to suggest new ways to do concept extraction would particularly strengthen the paper. Could the authors comment, at least as future work, how they could leverage their framework to evaluate possible new concept extraction methods?
2. (L. 128) Are the authors sure about the expressivity claim of K-means based methods? wouldn't expressity depend on the number of cluster K?
3. (L. 135) This seems a drawback of doing class-based concept extraction, rather than a problem specific to PCA. For example, how would K-means be able to detect the dog-head concept, if all the samples in the dog class have the dog-head concept?
4. The authors claim that NMF strikes an harmonious balance between K-Means and PCA. This somewhat should imply that NMF is a better choice. Could the authors elaborate on this? This would help strengthen the utility of their framework. The only argument is the claim that (L. 153) NMF is known to yield human-interpretable concepts. Is this not the case for PCA or K-Means? In the referenced papers, I cannot see a study on the utility of NMF concepts compared to PCA concepts.
5. As an extension to the above, it would be interesting to leverage the authors framework to more systematically pinpoint the properties of a certain dictionary learning method that make the extracted concepts more or less interpretable from a user perspective. The authors can perhaps expand on this in a future work section.
6. Evaluation of concept extraction: an additional evaluation could be performed by applying concept extraction methods to a model with ground truth concepts, e.g. a concept bottleneck model which was specifically trained with user-defined concepts as additional training signal.
7. As mentioned above, the results (especially the theorems) are unsurprising/trivial. but I will concede that I might be wrong, if the authors could further elaborate on the differences between their results and Ancona's. As observed by the authors, the last classification layer is by construction linear wrt the penultimate layer. Since the dictionary learning view essentially reveals that the concepts extracted at the penultimate layer are essentially a linear projection on a different base of the penultimate layer latent space. Therefore, it is a trivial extension of Ancona et al.'s work, the fact that all attribution methods are essentially equivalent on linear classifiers, and furthermore, achieve optimal fidelity, i.e. correlation of 1, which again is unsurprising for linearly correlated variables. Consequently, it seems also trivial to me that C-deletion and C-insertion also achieve optimal values since insertion and deletion greedy strategies are trivially optimal for positively monotonically correlated variables. In essence, I would say that Theorem 3.2 is an extension of Ancona's work, while Theorem 3.1 is essentially a corollary  to 3.2.
8. As additional comment to the point above, it seems also unfair to evaluate different layers using these definitions of deletion and insertions since they would correctly evaluate only layers that are linearly related to the network outputs.

**Limitations:**

Authors do not discuss or comment about possible limitations of their framework and possible future work. In particular, can their framework be used to derive new concept attribution scores and relative evaluation metrics, in case somebody might be constrained to not use the penultimate layer?

---

> ### Author Rebuttal · Authors · 2023-08-08
>
> We thank the reviewer for carefully reading and providing a constructive review with relevant remarks and suggestions that will enhance the quality of the paper.
>
> Please find below a point-by-point answer:
> * **About the lack of analysis of the Strategic Clustering graph:** The main goal of this article is to provide a theoretical framework to unify all concept-based methods. The strategic cluster graph was just proposed at the end as a use-case example and is secondary compared to the theoretical framework (which explains indeed the limited materials concerning those graphs). Further work, specifically focusing on strategic cluster graphs is in preparation.
>
> * **About the fact that the authors do not propose new concept extraction methods:** In this article, we have laid down the theoretical foundation, and proposed a series of benchmarks so that it becomes easier to propose and evaluate new concept extraction and attribution methods. We believe that this contribution is novel and important enough to deserve an entire article. Note that this opinion is also shared and supported by all other reviewers. Nonetheless, we agree with you that a natural follow-up of this article would be to investigate new methods (especially concept extraction methods). We actually plan to extend this article into a journal version where we aim to further investigate the topic by exploring probabilistic approaches such as Gaussian mixture and LDA. We appreciate this suggestion, as it aligns with our inclination to continue in this direction. In the meantime, we agree with you and have added a future work section that details these plans. Thank you for your valuable feedback and contribution to the improvement of the paper.
>
> * **About the unreferenced definition (L. 273)...** We have corrected these references. Thank you.
>
> * **Reviewer: About the expressivity claim of the K-Means (L. 128)**. The term expressivity was indeed poorly chosen, and we have now modified it. The intention was to highlight that for the reconstruction, a point could only belong to one cluster, which led to the inappropriate use of the term "lack of expressiveness." We have replaced this sentence by referring to the absence of compositionality to better convey the concept. Thank you for pointing out this improvement, and we appreciate your valuable feedback.
>
> * **Reviewer: (L. 135) This seems a drawback of doing class-based concept extraction, rather than a problem specific to PCA. For example, how would K-means be able to detect the dog-head concept, if all the samples in the dog class have the dog-head concept?** We respectfully disagree. The goal of a PCA is to find the directions that maximize the variance, therefore it will tend to omit features that are common to all samples (as the dog-head features for samples representing dogs). On the other hand K-means (on the features) aims at finding centroids that minimize the distance to features’ sample. Therefore, if the dog-head is a prevalent feature, it is likely that the K-Means will represent this feature. Interestingly, in the specific case of K-means with only one concept, the average indeed becomes the centroid. However, the issue with PCA arises from the absorption of the mean, which is a well-known fact. Thank you for clarifying this point, and we appreciate your valuable input.
>
> * **About the lack of argument on the utility of the NMF methods.** With all due respect, we believe that there is a misleading interpretation of our work : our arguments in favor of the NMF are bolstered by both theoretical and empirical arguments.  While the reviewer suggests that the sole claim hinges solely on NMF yielding human-interpretable concepts, we want to emphasize that we provide multiple theoretical arguments in Section 3.1 (l135, l139, l143, l147, l150, l154, l155) and empirical evidence in the 'Evaluation of concept extraction' (l178, l181, l189, l205, l206) section. We believe that these supporting elements validate the significance and validity of our conclusion. In addition, we wish to highlight that the referenced paper that studies the NMF utility is actually comparing it to PCA and K-means (see Fig 3 and Table 2 in [20]).
>
> * **About systematically pinpointing the properties of different dictionary learning method.** Thank you for this excellent suggestion. We have indeed added a discussion on this topic in the future work section.
>
> * **Reviewer: The results (especially the theorems) are unsurprising/trivial given de work of Anconda ?** We respectfully disagree. We are well aware of the really interesting  works of Ancona. There is indeed some similarities with the Anconda work, but our propositions (on linear optimality) differ for 2 main reasons : i) Ancona does not analyze Gradient, Smoothgrad, VarGrad, SquareGrad, RISE, etc. and ii)  Ancona does not provide any proof concerning mu-fidelity, C-deletion, or C-insertion, which is why we have introduced our theoretical elements. Note that the Mu-Fidelity metric (our) is a relaxation of the sensitivity-n metric (Ancona), therefore our proposition encompasses the Ancona's one. In addition, our work is also extending the linear optimality with respect to 2 other new metrics (C-Deletion and C-Insertion) that are not related to the Ancona metric. We sincerely hope that our response serves to alleviate any misconceptions and foster a clearer understanding of our intentions. Our aspiration is for the reviewer to consider the points we've presented, potentially leading to a reassessment of the evaluation rating. Your feedback is valuable to us, and we genuinely appreciate your time and consideration.

---

> > ### Comment · Reviewer_hQqh · 2023-08-18
> > **Thanks for the rebuttal!**
> >
> > I thank the authors for the nice rebuttal, and I apologize for the delay in coming back.
> >
> > Thanks for addressing most of my concerns. For now, I am increasing my score to 5 since
> > - indeed, the authors did provide some insights in comparing the 3 different methods within their framework
> > - I really like the strategic cluster graph/visualizations
> >
> > However, I do still have a concern about the theoretical part of the paper in that I think it is still a very marginal contribution on top of the similar work from Ancona et al.
> >
> > i) "Ancona does not analyze Gradient, Smoothgrad, VarGrad, SquareGrad, RISE, etc"
> >
> > Sure, I agree. However, I am fairly confident that these methods, as function of the gradient can be formulated in the (unifying) framework defined by Ancona et al. (happy to be proven wrong though). And as far as I can see in both the main paper and the supplementary, the results are stated only for Gradient*Input, Integrated Gradients, Occlusion and RISE, which are all addressed by Ancona (well, apart from RISE, but it's closely related to Occlusion).
> >
> > ii) "Ancona does not provide any proof concerning mu-fidelity, C-deletion, or C-insertion"
> >
> > Not directly, but my point is that it is not needed. While it's true, that "the Mu-Fidelity metric (our) is a relaxation of the sensitivity-n metric (Ancona)", it is not necessarily true that "our proposition encompasses the Ancona's one". Indeed, by restricting your result to the penultimate layer (which is linear wrt to the output), you are restricting your result exactly to the (if-and-only-if) conditions of Ancona's result. If you were to extend the result, than I would agree. but at this point not really.
> >
> > iii) "In addition, our work is also extending the linear optimality with respect to 2 other new metrics (C-Deletion and C-Insertion)"
> >
> > Similar to before, my argument is that by restricting the proof to the penultimate layer, I believe it becomes a lil bit uninsightful. By definition, if concept importances are perfectly positively correlated to the output (that is having a mu-fidelity of 1), then (I might be missing something, but) it seems quite trivial to me that C-Deletion and C-insertion are gonna be optimal

---

### Official Review · Reviewer_d9UA · 2023-07-06

**Soundness:** 3 good
**Presentation:** 3 good
**Contribution:** 3 good
**Rating:** 7
**Confidence:** 3

**Summary:**

This paper presents a novel framework for unifying existing post-hoc concept-based explainability methods in a cohesive, formalized form. Such a framework enables a principled study of key properties of state-of-the-art concept-based explainability methods, such as ACE, T-CAV, CRAFT, and ICE, and allows the authors to easily compare and contrast the quality of explanations generated by all of these approaches. To this end, this paper further introduces a large set of novel evaluation metrics for the quality of concept explanations together with a new simple framework to generate and visualize local concept-based explanations for a given prediction. These metrics are used to evaluate a variety of architectures in ImageNet and conclude that NMF-based concept-extraction approaches have significant potential compared to their clustering and PCA-based counterparts. Finally, the paper provides theoretical guarantees regarding the optimality of a set of concept-based explainability concept attribution methods, with respect to the fidelity and completeness of their explanations, when they are used to explain the latent space of the last layer of a DNN.

**Strengths:**

Thank you for this very interesting and amazingly-crafted work; I thoroughly enjoyed reading it. I believe this paper’s goal of introducing a unifying formal framework to study most concept-based XAI methods is very important in a field lacking principled and theoretical analysis. Overall, I believe that this paper’s main strengths are:

1. This paper’s core motivation, the framework it introduces, and the metrics it proposes are all worthy of study and of interest to the wider XAI community. These can become particularly handy for other researchers to evaluate, compare, and study novel methods against existing ones.
2. Similarly, the key insights developed by this paper, in particular the optimality theorems and their surprising results on NMF-based methods, are all novel and potential sources for interesting future research.
3. The paper is extremely well-written, with the writing being very clear and the figures nicely complementing the rest of the text. In particular, I like the non-standard approach to writing this work (not strictly following the now-boring intro-methods-results-discussion pipeline) and think it is worth pointing out.
4. There is a very good mix of theoretical results and interesting insights from very simple experiments. In particular, I found the use of matroids for the proofs of the proposed theorems to be very interesting.
5. The introduction of a framework that enables both global and local concept-based explanations with one method is novel and particularly useful in practice.

**Weaknesses:**

Overall, I think this paper introduces a very useful framework and set of metrics that may be of real interest to the wider community. I am leaning towards acceptance for this reason. However, I have some concerns/questions regarding some specific bits of the paper as well as some possible gaps in this work.  Specifically, I believe the following are that paper's main limitations:

1. Some key and more recent methods are not mentioned or cited in this work. These include Concept Gradients [1] (which I am not sure if it directly fits in the proposed framework, although this would be a good test for the proposed ideas) and, more importantly, as this is extremely related to how the reconstruction of the latent space is framed in this work, completeness-aware concept discovery (CCD) [2]. Similarly, traditional topic models, such as latent Dirichlet allocation (LDA), are highly related to some of the ideas discussed in the concept extraction part of this framework.
2. I am still uncertain whether ACE and T-CAV, as originally proposed in their original papers, can be expressed using this paper’s framework. If so, this weakens this paper's unification claim, a key contribution according to the authors. This may be a misunderstanding on my end, but please take a look at my questions below to possibly clarify this.
3. Against general convention, none of the figures and tables include error bars, making it harder to understand when differences are statistically significant.
4. The paper is missing key details regarding their empirical evaluation. These include model sources, licenses, and references showing the training setups for the original pre-trained models if models were not trained from scratch.

### References

[1] Bai, Andrew, et al. "Concept Gradient: Concept-based Interpretation Without Linear Assumption." *The Eleventh International Conference on Learning Representations*. 2022.

[2] Yeh, Chih-Kuan, et al. "On completeness-aware concept-based explanations in deep neural networks." *Advances in neural information processing systems* 33 (2020): 20554-20565.

**Questions:**

Regarding the weaknesses discussed above, I will gladly consider increasing my score if some of these concerns are appropriately addressed by the authors (especially if they come from a misunderstanding on my end). To help with this discussion, I believe the following questions could be a good starting point:

1. Could you please elaborate on how CCD fits within your framework and whether it would be feasible to include a topic-modelling-based concept extractor as part of the comparison in section 3.1? The lack of discussion of this method, a key algorithm in unsupervised concept learning, weakens the universal unification argument in this paper.
2. One of the main claims of the paper is that the framework unifies “all modern concept-based explainability methods”. Yet, I fail to see how it can explicitly describe T-CAV in the form originally proposed by their authors. Could you please clarify how T-CAV , as defined in its original paper, can be explicitly framed in the proposed framework? I can see how it can be framed if one assumes that the latent space from which we extract the CAVs from can be explicitly reconstructed from all the concepts we learn (and is done so as part of inference). This, however, is in no way an assumption of the original T-CAV process and, in fact, it is a very strong constraint that in practice fails to apply. This is because we will most certainly not have all the necessary concepts for perfect reconstruction for most tasks.
3. Regarding how ACE is framed in the proposed framework, I am still a bit doubtful that it is such a straight fit as claimed by this work. If I recall correctly, ACE uses clustering just to learn subsets of concept examples from which CAVs can later be learnt using linear separators. This paper, however, seems to suggest that ACE uses the cluster centroids themselves as the CAVs. If my understanding of that claim is correct, then I believe that’s wrong. Could you please clarify this point? If my understanding is correct, then this would require correctly updating a text where it claims that ACE falls within the clustering concept extractor family of methods.

Regarding suggestions for the presentation, the paper is very well-written (thank you so much!). Nevertheless, the following possible errors and typos could be addressed before a camera-ready version:

- **[important]** The appendices referenced in the main body of the paper seem not to be correctly aligned with their correct sections in the submitted supplementary material.
- The use of bold symbols is a bit confusing. Sometimes it is used for tensors/vectors (e.g., $X$ and $x$ in line 93), sometimes it is used for functions even when their output is not a vector (e.g., $g$ in line 98), and sometimes it is used to refer to a scalar (e.g., line 509 in the appendix). Traditionally it is just used for vectors/tensors and not functions names or scalars. It is ok if this does not follow tradition, but it could benefit from consistency (as, say, some scalars are not bolded as the bias term $b$ sometimes).
- Typo in line 112: it should probably be “This allows us…” rather than “It allows us…” as the “it” is unclear in that context.
- In line 177 it should probably be “the best score” rather than “the highest score”  as PCA achieves the best score amongst all methods (which is the lowest, not the highest for relative $\ell_2$.
- The use of mix cases in “The Last Layer as a Promising Direction” in line 212 is inconsistent with the rest of the subtitles.
- In line 214, a full stop “.” is missing before “To explore”.
- The caption of Table 1 should be on top of the table to follow NeurIPS standards.
- Line 220-221 read a bit off. Maybe a typo or a word is missing there.
- Figure 3’s x-axis should be “# of Concepts” rather than “# of Concept”
- In that same figure, it would be beneficial to use different colors or markers as it is otherwise hard to disentangle some of the methods (and then even harder to find the respective color in Figure 4 without the legend right there).
- CAT is redefined in line 265.
- In line 279 “zero” should be “zeros”.
- Line 296 is missing “are” between $y \in \mathbb{R}$ and “the logits…”
- In line 339, “this” should be “these”
- nit: It is uncommon to have more than one paragraph in the abstract. Just pointing this out in case this comes from an accidental new line inserted in the abstract.

**Limitations:**

This work does not discuss any limitations of the proposed framework, metrics, or results. It could certainly benefit from a short section where it is discussed how this framework may fail to generalize to some existing concept-based methods and how it could possibly be expanded in the future. Similarly, the manuscript does not discuss possible societal impacts of the work. However, I believe no clear negative societal impacts could arise from this paper's contributions and methods.

---

> ### Author Rebuttal · Authors · 2023-08-08
>
> We are delighted to learn that the reviewer found the paper enjoyable and useful for the community. We extend our sincere gratitude to the reviewer for investing time and effort in providing a thorough review!
>
> Please find below a point-by-point answer:
>
> * **Reviewer: Some key and more recent methods are not mentioned or cited in this work. These include Concept Gradients [1] (which I am not sure if it directly fits in the proposed framework, although this would be a good test for the proposed ideas) and, more importantly, as this is extremely related to how the reconstruction of the latent space is framed in this work, completeness-aware concept discovery (CCD) [2]. Similarly, traditional topic models, such as latent Dirichlet allocation (LDA), are highly related to some of the ideas discussed in the concept extraction part of this framework.** The authors were not aware of the CCD work, and we appreciate your observation. As you rightly pointed out, we have now added a discussion of CCD in the appropriate section. Regarding LDA, we acknowledge its potential as a good solution (also noted by reviewer  hQqh), and we believe it could serve as a promising starting point for future research on topic-modelling-based concept extraction.
>
> * **Reviewer: I am still uncertain whether ACE and T-CAV, as originally proposed in their original papers, can be expressed using this paper’s framework. If so, this weakens this paper's unification claim, a key contribution according to the authors. This may be a misunderstanding on my end, but please take a look at my questions below to possibly clarify this. One of the main claims of the paper is that the framework unifies “all modern concept-based explainability methods”. Yet, I fail to see how it can explicitly describe T-CAV in the form originally proposed by their authors. Could you please clarify how T-CAV , as defined in its original paper, can be explicitly framed in the proposed framework? I can see how it can be framed if one assumes that the latent space from which we extract the CAVs from can be explicitly reconstructed from all the concepts we learn (and is done so as part of inference). This, however, is in no way an assumption of the original T-CAV process and, in fact, it is a very strong constraint that in practice fails to apply. This is because we will most certainly not have all the necessary concepts for perfect reconstruction for most tasks.** T-CAV (from [18]),  is a sensitivity score that quantifies how important a concept is (a CAV) to a classification score. In the original paper, it is computed using a gradient projected on the CAV (i.e. the directional directive). We have shown in section 3.2., that the T-CAV sensitivity score is equivalent to the gradient of the activation reconstruction (i.e $g(uV^{T})$) w.r.t to the representation (i.e. $u$). Said differently, the T-CAV score is a particular CAT (concept attribution method) that leverages the gradient method. Note that in the original T-CAV article [18], the CAVs are extracted using human judgment (it is not an automatic concept extraction method), therefore we could not fit this non-automated concept extraction method in our framework. Concerning ACE, both the attribution and the concept extraction fits in our framework. The attribution method of ACE is a T-CAV (i.e. the gradient Concept ATtribution) and its concept extraction method is based on clustering. Note that the centroids themselves are not the CAVs in ACE, but the training of CAVs is based on the distances to the centroids. This strongly suggests that the vector orthogonal to the decision boundary points towards or is close to the centroid. However, we agree with you and have added this clarification to the paper. Thank you for bringing up this important detail, and we appreciate your input.
>
> * **Reviewer: The paper is missing key details regarding their empirical evaluation. These include model sources, licenses, and references showing the training setups for the original pre-trained models if models were not trained from scratch.**
> We have included these details in a dedicated section in the appendix. The models used are pre-trained models on ImageNet from the Keras library, and we have provided the configurations of these models.
>
> * **Reviewer: This work does not discuss any limitations of the proposed framework, metrics, or results. It could certainly benefit from a short section where it is discussed how this framework may fail to generalize to some existing concept-based methods and how it could possibly be expanded in the future. Similarly, the manuscript does not discuss possible societal impacts of the work. However, I believe no clear negative societal impacts could arise from this paper's contributions and methods.** Another reviewer requested a limitation section, and as per their suggestion, we have included it in the appendix.
>
> * **Concerning the minor error and the typos** We appreciate the corrections regarding the typos, and we have taken them into account and made the necessary adjustments accordingly. As a result, the manuscript has been improved.

---

> > ### Comment · Reviewer_d9UA · 2023-08-15
> >
> > Dear Authors,
> >
> > Thank you for your careful rebuttal and for replying to my questions/concerns regarding this work. As discussed in my review, and as echoed by the rebuttal, it is my understanding that T-CAV and ACE, as originally framed in their original works (which include the CAV discovery part), cannot be fully framed using the proposed framework. I agree with the authors that after the CAVs have been extracted, this framework allows one to capture both of these methods. However, this is a weaker claim than the one made in this paper and damages a bit the "unifying" part of this work's main contribution. Similarly, with my concern regarding CCD/LDA, I would've appreciated it if the authors could have described how this method fits within this framework, say by providing a glimpse of what will be added about these two methods in the updated manuscript.
> >
> > Given that my main concerns/questions with this work have not yet been fully addressed, I am leaning towards maintaining my original score. While I honestly believe there is value in this paper's work, the missing details, the slightly too strong unifying claims, and the lack of positioning with CCD/LDA (two well-known methods in this area), all lead me to maintain my current position. Still, I want to clarify that my position is more towards (weak) acceptance side as I think there is real value here, which, if positioned more accurately by considering the feedback from all reviewers, could be useful for the community.

---

> > > ### Author Response · Authors · 2023-08-18
> > > **Response to d9UA (1/2)**
> > >
> > > We thank the reviewer for the reply.
> > >
> > > * Concerning ACE : It is an automatic concept extraction method that fits the framework we have proposed. ACE defines the CAVs as the centroids clusters extracted using k-means (see Fig 1b in [19]. Such extraction methods fits our framework (see eq 1 and also line 123-125). Once the CAVs are extracted, the ACE authors uses the T-CAV method to evaluate the concept importance. As demonstrated in our article (see Eq. 4), the T-CAV method could be expressed as a particular Concept ATtribution method (CATs, see section 3.2). Therefore, both the concept extraction method, and concept importance score of ACE enter in our our framework
> > >
> > > * Concerning the T-CAV article: I would like to emphasize than in [18], the authors **DO NOT** introduce an automatic concept extraction method, as they label the concept (i.e. the CAVs) using human annotation (see Fig 1a in [18]). Our claim is to unify automatic concept extraction methods, <ins>so the concept extraction part of [18] is not in the scope of our claim</ins>. However once the CAVs are extracted, they use (and introduce) the T-CAV method to assess the importance of concepts. the T-CAV method fully fits the concept scoring part of our framework (see section 3.2 and Eq 4).
> > >
> > > * The link between the LDA and our framework is indeed not straightforward, mostly because we haven't adopted a probabilistic formulation of the dictionary learning step (it is a linear algebra formulation). Note that our dictionary learning formulation could be further generalized leveraging probabilistic formulation:
> > > $$ U^{\star}, V^{\star} = \arg \max_{U,V} \log p(A|U) \quad \textnormal{s.t.} \quad p\sim\mathcal{N}(f(U),I) \quad \textnormal{with} \quad f(U) = UV^{T}$$
> > > This formulation allows to extend the likelihood probability ($p(\cdot|U)$) to
> > > to other distributions (other than Gaussian).  In this frame, the LDA just becomes a case where we impose concepts (or topics in the case of text) to be dependent on a Dirichlet distribution (with learnable parameters). Note also that previous works have shown that one could interpret the LDA as a mixture of multinomial PCA (see [A, B]), and the PCA method falls within our framework (see section). Although this probabilistic formulation is interesting, we think that our 'linear algebra' formulation is easier to grasp and give better intuition. We propose to enrich the perspective section to mention the probabilistic formulation of the dictionary learning problem so that we can frame the LDA method.
> > >
> > > [A] : Buntine, Wray. "Variational extensions to EM and multinomial PCA." _European Conference on Machine Learning_. Berlin, Heidelberg: Springer Berlin Heidelberg, 2002. \
> > > [B] : Hoffman, Matthew, Francis Bach, and David Blei. "Online learning for latent dirichlet allocation." _advances in neural information processing systems_ 23 (2010). \

---

> > > ### Author Response · Authors · 2023-08-18
> > > **Response to d9UA (2/2)**
> > >
> > > Here is the rest of our answer :
> > >
> > > * Concerning CCD (Completeness-aware Concept Based Explanation) from [C].  The ConceptSHAP method also falls into the framework of dictionary learning, through the more general lens of parameters estimation. We recall their formulation below:
> > > $$\arg\max_{v,d}\log\mathbb{P}[g_y(d(u_v(x)))]+R(v)$$
> > > We recognize the joint learning of the concepts $v$ , and  the coefficients $u_v$. Contrary to PCA, NMF or K-Means that solely minimizes the reconstruction error in activation space (regardless of the downstream task), here the reconstruction error is monitored through the predictions $g_y(g(v_c(x)))$ on the final task. Therefore, a more general formulation would be:
> > > $$\arg\min_{d,v,u}D((x,y), h(x), d(v,u))+R(v,u)$$
> > > where:
> > > * $(x,y)$ is the dataset with (input, label) pairs, $h$ the feature extractor of the neural network $f=g\circ h$, and $h(x)$ the activations.
> > > * $d$ (like *decode*) is a function that reconstructs activations from the concepts $v$ and the scores $u$
> > > * $D((x,y), \cdot, \cdot)$ is an arbitrary divergence measure, possibly conditioned on the dataset
> > > $R$ a regularization term on the concepts/coefficients. If we allow indicator functions in $R$ (i.e that are either $0$ or $+\infty$ depending on a boolean condition) this framework also allows us to incorporate the constraints.
> > >
> > > In the case of PCA/NMF/K-Means, the divergence $D((x,y), g, h, \cdot)$ is the squared Frobenius norm between reconstructed activations and the original’s ones, and $d(U,V)=UV^T$.  For NMF, the regularization is the indicator function $R(c,v)=1_{c\geq 0,v\geq 0}$ which ensures the positiveness.
> > >
> > > For CCD, $D((x,y),\cdot,\cdot)$ is related to the log likelihood $\log\mathbb{P}(g_y(\cdot))$, $d$ is the MLP that maps coefficients $u$ and concepts $v$ to reconstructed activations (they call it $g$ in their paper, not to be confused with our $g$), while $R$ is their regularization based on K-Means.  We recall that the loss $\log\mathbb{P}(g_y(\cdot))$ can be rewritten as a cross-entropy between predictions and one-hot encoding of the labels, and furthermore cross-entropy is itself a divergence (the KL divergence to be more precise) with an additional entropic regularization term. Therefore we have:
> > > $$D((x,y), g, h,  d(u, v))=KL(g(d(u, v)), \text{one-hot}(y))+\text{entropy}(g(d(u, v)))$$
> > >
> > > In summary, the formulation $\arg\min_{d,v,u}D((x,y), h(x), d(v,u))+R(v,u)$ is more general than ours and allow to express all existing methods (even LDA). We thanks the reviewer to bring this interesting point to our attention, and we plan to use the extra page (if the paper is accepted) to mention all those different formulations of our framework (probabilistic and Divergence like).
> > >
> > >
> > > [C] Yeh, Chih-Kuan, et al. "On completeness-aware concept-based explanations in deep neural networks." Advances in neural information processing systems 33 (2020): 20554-20565.

---

> > > > ### Comment · Reviewer_d9UA · 2023-08-18
> > > > **Updating review score**
> > > >
> > > > Dear Authors,
> > > >
> > > > Thank you for outlining all these details and for the promise to include this discussion as part of the updated manuscript. I believe such a discussion and the reformulation of your framework as described above will add a lot of strength to your paper. More importantly, I find the arguments presented above both interesting and well-positioned within their respective works (it certainly clarified some of my concerns/doubts). Given this and the rest of your rebuttal, I am increasing my score to an accept (7). Best of luck!

---

### Official Review · Reviewer_Y6po · 2023-07-07

**Soundness:** 4 excellent
**Presentation:** 4 excellent
**Contribution:** 3 good
**Rating:** 7
**Confidence:** 3

**Summary:**

In this paper, the authors propose a unified perspective on post-hoc concept-based explanation methods. The main intuition underlying the work revolves around the fact that typical concept-based explanations can be considered as a two-stage process whereby, initially, a concept vocabulary is learned and, lastly, each concept's importance is evaluated. In light of their theoretical analysis, the authors describe a variegated class of concept-based explanations, showing in particular that the penultimate layer is particularly suited for concept analysis. In addition, the authors derive that some concept-importance methods are theoretically sound in the penultimate layer.
Concerning the experimental investigation, the authors propose a set of metrics for comparing different explainability methods in both the settings of concept dictionary learning and concept importance estimation. The experiments are specifically tailored to evidence differences between existing methods.
The authors also contribute to a dedicated interface for visualizing the clusters graph on the ImageNet classes.

**Strengths:**

In this paper, the theoretical analysis is extensive and well curated and the presentation is clear and easy to follow. The main strength of the paper is introducing a unified framework for analyzing concept-based explanations, which reveals particularly suited to address the pros and cons limitations of existing methods. In particular, the linear decomposition of the activation matrix in the concept vectors covers several existing approaches. This is of particular help also for proceeding with the research in the field.

The integration of three new metrics for concept extraction helps to compare several strategies, among which NMF decomposition seems preferable. Similarly, the authors propose three metrics to evaluate concept importance methods, deriving theoretical guarantees for 4 methods in the penultimate layer. This strengthens the idea that concept analysis is best when conducted in the deepest layer of neural networks. The experiments are illustrative to provide a discrimination between different methods and complete the theoretical analysis conducted.

I also found helpful the introduction of the strategic cluster graph, which provides an intuitive interface for visualizing concept-level explanations of the model.

**Weaknesses:**

I have no major concerns about the presented material, which is very curated and high quality.

Nonetheless, I find two weaknesses on the general method proposed. First of all, learning a dictionary of concepts in unsupervised manner does not give any guarantee that the extracted concepts do actually possess a well-defined semantics. This requires a verification with an end user, as for example in [1], and, so, the considered methods for post-hoc concepts analysis are limited and do not ensure interpretability.

Another weakness amounts to extracting the concepts in a linear manner and it seems that classes are actually related to concepts in a kind of AND rule. This does not hold in general, where classes might depend on non-linear relations between concepts. As an example, take a class resulting from a XOR operation between two concepts, there is no way to extract this information with these methods.

I hope to see some discussion on these points.

[1] Biswas, Shreyan, et al. "CHIME: Causal Human-in-the-Loop Model Explanations." Proceedings of the AAAI Conference on Human Computation and Crowdsourcing. Vol. 10. No. 1. 2022.

**Questions:**

Does the analysis offer guidance toward non-linear cases? Is it possible to extend the current approach to that scenario?



**Limitations:**

The interpretability of concepts is not discussed at length; this addition may strengthen the authors' findings and provide guidance to improve the research on concept-based explanations.

---

> ### Author Rebuttal · Authors · 2023-08-08
>
> We thank the reviewer for the very nice review. Once again, we extend our appreciation for the time and the very comprehensive review.
>
> Please find below a point-by-point answer:
>
> * **Reviewer : 'Learning a dictionary of concepts in an unsupervised manner does not give any guarantee that the extracted concepts do actually possess a well-defined semantics. This requires a verification with an end user, as for example in [1], and, so, the considered methods for post-hoc concepts analysis are limited and do not ensure interpretability'.** We wish to highlight the fact that the dictionary learning technique is applied on images belonging to the same class (which might be interpreted as a ‘weak’ form or supervision). Therefore we have the guarantee that the learned concepts are well describing a given class. Nonetheless, we agree with the reviewer that there is no guarantee that the learned concepts are semantically meaningful. However, [20] has demonstrated that, in practice, the extracted concepts tend to align with human judgment (especially when extracted using an NMF).
> We have expanded a section on limitations to address this point.
>
> * **Reviewer : 'Another weakness amounts to extracting the concepts in a linear manner and it seems that classes are actually related to concepts in a kind of AND rule. This does not hold in general, where classes might depend on non-linear relations between concepts. As an example, take a class resulting from a XOR operation between two concepts, there is no way to extract this information with these methods.'**
> The main advantage of imposing a linear decomposition is that the resulting concepts are more easily interpretable. We refer the reviewer to [18] for an interesting discussion about linear interpretability. However we agree with the reviewer that such a constraint limits the expressivity of the learned concepts. We have expanded the limitation section to better highlight this point.
>
> * **Reviewer : 'Does the analysis offer guidance toward non-linear cases? Is it possible to extend the current approach to that scenario?**
> Indeed, the limitation you mentioned has been highlighted several times. As for the proof regarding C-Deletion and C-Insertion, we could potentially extend it using matroid theory to cover cases of submodular functions. However, it seems challenging to prove anything of interest in a non-linear case without additional assumptions.'

---

### Official Review · Reviewer_xA3z · 2023-07-10

**Soundness:** 3 good
**Presentation:** 4 excellent
**Contribution:** 3 good
**Rating:** 7
**Confidence:** 5

**Summary:**

The paper provides a unifying perspective for the concept extraction / importance estimation method via the dictionary learning framework. Namely, depending on the constraints on $U$ and $V$, the framework includes ACE, PCA, CRAFT and ICE as special cases. Then, the paper compares three difference approaches, PCA / NMF / K-Means, for the dictionary learning framework using several metrics. Moreover, by generalizing the attribution methods in XAI, the concept importance values are also computed and compared across different attribution methods for the NMF concept extraction. The theory for the concepts extracted from the penultimate layer is also given. Finally, by defining reliability and prevalence of the concepts, the visual and qualitative results are shown, which are compelling.

**Strengths:**

- Unifying effort for the concept extraction / importance is sound and the theoretical results back the approach of the paper.
- Combining the attribution method with concept extraction is interesting and the reliability / prevalence notions make sense.
- The visualization results are compelling.


**Weaknesses:**

- I think it would be beneficial to show the failure case of the proposed method. Figure 1/2 show compelling results for three classes (Espresso, Zucchini, and Lemon) but would there be classes that the proposed method fails to give meaningful explanation? What would those classes be like? It would be informative to show such examples as well so that people who use the proposed technique could be cautious.
- It seems like the local importance score depends on the entire test set -- namely depending on what other images exist in the test set, the reliability and prevalence should vary. What are the variance of those values depending on the test set? Maybe cross-validation like evaluation should be meaningful.


**Questions:**

- The strategic graph is obtained by the Integrated Gradient attribution method -- how would be different across different attribution method?
- How would the strategic graph change for different test sets?


**Limitations:**

Please see the comment for weakness.

---

> ### Author Rebuttal · Authors · 2023-08-08
>
> We thank the reviewer for the time and the suggestions in the reviews.
>
> Please find below a point-by-point answer:
> * **Reviewer : 'I think it would be beneficial to show the failure case of the proposed method. Figure 1/2 show compelling results for three classes (Espresso, Zucchini, and Lemon) but would there be classes that the proposed method fails to give meaningful explanation? What would those classes be like? It would be informative to show such examples as well so that people who use the proposed technique could be cautious.'**
> To be entirely honest, we have not encountered any cases where the method "does not work." As evidence, we have included  a website containing the 1000 classes of ImageNet (https://anon1285.github.io/neurips1285/)
> However, you are correct, and as requested by several reviewers, we will create a dedicated section on limitations of the framework.
>
>
> * **Reviewer : 'It seems like the local importance score depends on the entire test set -- namely depending on what other images exist in the test set, the reliability and prevalence should vary. What are the variance of those values depending on the test set? Maybe cross-validation like evaluation should be meaningful. How would the strategic graph change for different test sets?'**
> The local importance score does not vary with the distribution as it is computed, for one single image, with the concept attribution score. However, you are correct for  the reliability and prevalence scores that depend on the aggregation of a set of points. Since the local importance for a point remains constant, the prevalence is exclusively influenced by the stability of the concept extraction method (i.e. its ability to give the same sets of concepts whatever the testing distribution), which we have measured in Figure 2, plot 5.  The prevalence we have computed is based on the NMF concept extraction method, which exhibits a good stability. Therefore, the prevalence is only slightly impacted by the distribution of the test images.
>
>
> * **Reviewer : 'The strategic graph is obtained by the Integrated Gradient attribution method -- how would be different across different attribution method?'**
> The strategic graph is computed using the last-layer representation, and we have demonstrated (in section “A note on the last layer”) that several attribution methods boil down to the same form during concept extraction on the last layer. Consequently, Integrated Gradient, Gradient Input, Rise, and Occlusion will yield the same strategic graph. Furthermore, we have demonstrated (theorem 3.1 and 3.2)  that these feature importance methods are optimal with respect to the three fidelity metrics. Hence, we exclude the other methods from consideration.

---

> > ### Comment · Reviewer_xA3z · 2023-08-18
> >
> > Thanks for the rebuttal, and they have mostly resolved my issues. I think the proposed work can provide a nice tool set for interpreting the neural network models, hence I raised my score to "Accept".

---

### Author Rebuttal · Authors · 2023-08-08


The reviewers found the paper "*extremely well-written*" "*thoroughly enjoyable to read*" and appreciating the "*non-standard approach*" taken. Regarding the theoretical work, you found it “*interesting and extensively curated*”, with reviewers finding the use of matroids in proofs "*very interesting*" and acknowledging the importance of introducing a formal framework for concept-based XAI methods in a field lacking theoretical analysis”. Furthermore, the paper receives positive feedback on its overall quality, with reviewers expressing gratitude for this "*very interesting and amazingly-crafted work*" and having "*no major concerns*" about its curated and high-quality content.

However, we acknowledge that you have raised certain points and identified weaknesses in the paper. During the rebuttal period, we meticulously examined the critiques provided by all four reviewers and have focused on addressing the following two main issues that have been highlighted:

1. The need for more examples of the analysis of the Strategic Clustering Graph and its potential limitations. We have included an anonymised website that showcases the Strategic Cluster Graph for the 1000 classes of ImageNet (https://anon1285.github.io/neurips1285/). Upon analysis, we did not identify any specific cases of failure.

2. The need to include a future work section, explaining the potential next steps, particularly in terms of applying the framework to discover/test new extraction methods. We acknowledge that this framework can not only be used to study existing methods but also to propose novel ones. We are excited about the possibility of pushing the boundaries using this framework. In this article, we have chosen to focus on establishing the foundation of the framework and have started exploring new extraction methods with it. We have added a dedicated section to discuss future research directions, and we sincerely appreciate your valuable suggestions.

Furthermore, we have diligently incorporated the necessary citations and clarifications into our Related Work and Discussion sections. We will now proceed to address each reviewer's remaining comments directly.

---

### Decision · Program_Chairs · 2023-09-21

**Decision:**

Accept (spotlight)

**Comment:**

This paper received four positive ratings: 7775. Authors support its contributions to unifying the concept extraction and importance and its value to the community, together with sound theoretical results and analyses. The concerns of reviewers are also well addressed. Overall, it is recommended to be accepted. And, based on the discussions with reviewers, more details and discussions need to be added in the final version.